# Signatures of transposon-mediated genome inflation, host specialization, and photoentrainment in *Entomophthora muscae* and allied entomophthoralean fungi

**Jason E Stajich[1]\*, Brian Lovett[2], Emily Lee[3†], Angie M Macias[4], Ann E Hajek[5], Benjamin L de Bivort[3], Matt T Kasson[4], Henrik H De Fine Licht[6], Carolyn Elya[3,7]\***

[1]Department of Microbiology and Plant Pathology, University of California-Riverside, Riverside, United States; [2]Emerging Pests and Pathogens Research Unit, USDA-ARS, Ithaca, United States; [3]Department of Organismic and Evolutionary Biology, Harvard University, Cambridge, United States; [4]Division of Plant and Soil Sciences, West Virginia University, Morgantown, United States; [5]Department of Entomology, Cornell University, Ithaca, United States; [6]Section for Organismal Biology, Department of Plant and Environmental Sciences, University of Copenhagen, Copenhagen, Denmark; [7]Department of Molecular and Cellular Biology, Harvard University, Cambridge, United States

**\*For correspondence:**
jason.stajich@ucr.edu (JES);
cnelya@gmail.com (CE)

**Present address:** [†]New York Genome Center, New York, United States

**Abstract** Despite over a century of observations, the obligate insect parasites within the order Entomophthorales remain poorly characterized at the genetic level. In this manuscript, we present a genome for a laboratory-tractable *Entomophthora muscae* isolate that infects fruit flies. Our *E. muscae* assembly is 1.03 Gb, consists of 7810 contigs and contains 81.3% complete fungal BUSCOs. Using a comparative approach with recent datasets from entomophthoralean fungi, we show that giant genomes are the norm within Entomophthoraceae owing to extensive, but not recent, Ty3 retrotransposon activity. In addition, we find that *E. muscae* and its closest allies possess genes that are likely homologs to the blue-light sensor *white-collar 1*, a *Neurospora crassa* gene that has a well-established role in maintaining circadian rhythms. We uncover evidence that *E. muscae* diverged from other entomophthoralean fungi by expansion of existing families, rather than loss of particular domains, and possesses a potentially unique suite of secreted catabolic enzymes, consistent with *E. muscae*'s species-specific, biotrophic lifestyle. Finally, we offer a head-to-head comparison of morphological and molecular data for species within the *E. muscae* species complex that support the need for taxonomic revision within this group. Altogether, we provide a genetic and molecular foundation that we hope will provide a platform for the continued study of the unique biology of entomophthoralean fungi.

## eLife assessment

This **important** study reports on the genome evolution of a poorly studied fungal group. By combining long-read sequencing and different bioinformatic analyses, the authors show that the giant genome of *Entomophthora muscae* expanded due to extensive transposable element activity. The strength of evidence is **compelling** and the authors are to be commended for their multiple comparative analyses of gene content along with transparently written and visualized techniques,

data curation, and methods. This paper will be of relevance to fungal biologists as well as to evolutionary biologists interested in the study of genome size dynamics.

## Introduction

Fungal insect pathogens play vital roles within ecosystems and have significant agricultural and economic impacts for human beings (*Lovett and St Leger, 2017*). While several ascomycete entomopathogens have been extensively studied (e.g. *Beauveria bassiana*, *Metarhizium anisopliae*, etc.), many others, especially those within the fungal phylum Zoopagomycota (formerly Zygomycota), have received comparatively little attention and are poorly understood. Zoopagomycota is the earliest diverging lineage of non-flagellated fungi (*Spatafora et al., 2016*) and consists of saprotrophs and animal and fungal parasites (*Figure 1A*). In particular, fungi within the order Entomophthorales (subphylum Entomophthoromycotina) are obligate and often highly specialized insect parasites that drive epizootic events and have massive impacts on local insect populations. For example, a 1989 outbreak of *Entomophaga maimaiga* was observed to cause population collapses of the invasive spongy moth, *Lymantria dispar*, across the northeastern US (*Hajek et al., 1990*). In 1983, an outbreak of *Entomophthora muscae* in the black dump fly (*Ophyra aenescens*) was found to affect nearly the entire population (*Mullens et al., 1987*). Owing to their deadliness for specific insect hosts, many entomophthoralean fungi have attracted attention with the hope of developing more targeted pesticides (*Pell et al., 2001*). Yet, little progress has been made, due in part to the enduring mysteries of their biology.

In addition to ecological importance and promise for agricultural applications, many entomophthoralean fungi change the behavior of their hosts so dramatically that infected individuals are often referred to as 'zombies' (*de Bekker et al., 2021*). For example, *Entomophthora muscae* drives infected fly hosts to climb a nearby surface (a phenomenon referred to as 'summit disease') (*Evans, 1989*; *Lovett et al., 2020b*; *Roy et al., 2006*), extend its proboscis and raise its wings immediately prior to death (*Elya et al., 2018*; *Krasnoff et al., 1995*). Likewise, *Entomophaga grylli* and *Pandora formicae* lead grasshoppers and ants to die clinging to or biting the tops of plants, respectively (*Boer, 2008*; *Marikovsky, 1962*; *Pickford and Riegert, 1964*). *Massospora* spp. employ more active modes of transmission by their living, manipulated cicada hosts (*Lovett et al., 2020a*), including hypersexual behaviors that increase the rate of contact transmission (*Cooley et al., 2018*).

A major challenge in understanding the biology of entomophthoralean fungi is the difficulty of culturing these fungi in the laboratory. While some species can be grown in vitro (*Freimoser et al., 2000*; *Grundschober et al., 1998*; *Hajek et al., 2012*; *Holdom, 1983*; *Hua and Feng, 2003*) or, less commonly, inside lab-reared insects (*Elya et al., 2018*; *Mullens, 1986*), such methods are only available for a few species. Genomes for these organisms have also been difficult to acquire owing to their extreme size, high proportion of repeat content, and the difficulty of obtaining sufficient high-quality, high-molecular-weight DNA (*Anna Muszewska, 2014*; *Stajich et al., 2022*). Here, we describe a long-read based genome assembly of the fly pathogen *Entomophthora muscae* isolated from fruit flies (strain *E. muscae* 'Berkeley', *Elya et al., 2018*). We used this genome in a comparative analysis with other available entomophthoralean transcriptomic and genomic datasets that addressed four main biological questions: (1) Why are the genomes so large? (2) What are patterns in predicted functionality across Entomophthorales? (3) What elements are unique to *E. muscae*? (4) Are phylogenetic determinations made using morphological characteristics consistent with those based on molecular data? In assembling the *E. muscae* genome and using it to address these initial questions, we hope to provide biological insight into the Entomophthorales as well as provoke additional exploration into these understudied fungi. The answers we uncovered each reveal new avenues of research that will lead to a better understanding of the evolution of entomophthoralean insect pathogens, particularly the influence of vastly proliferated transposable elements, in support of future work and applied use of these insect pathogens.

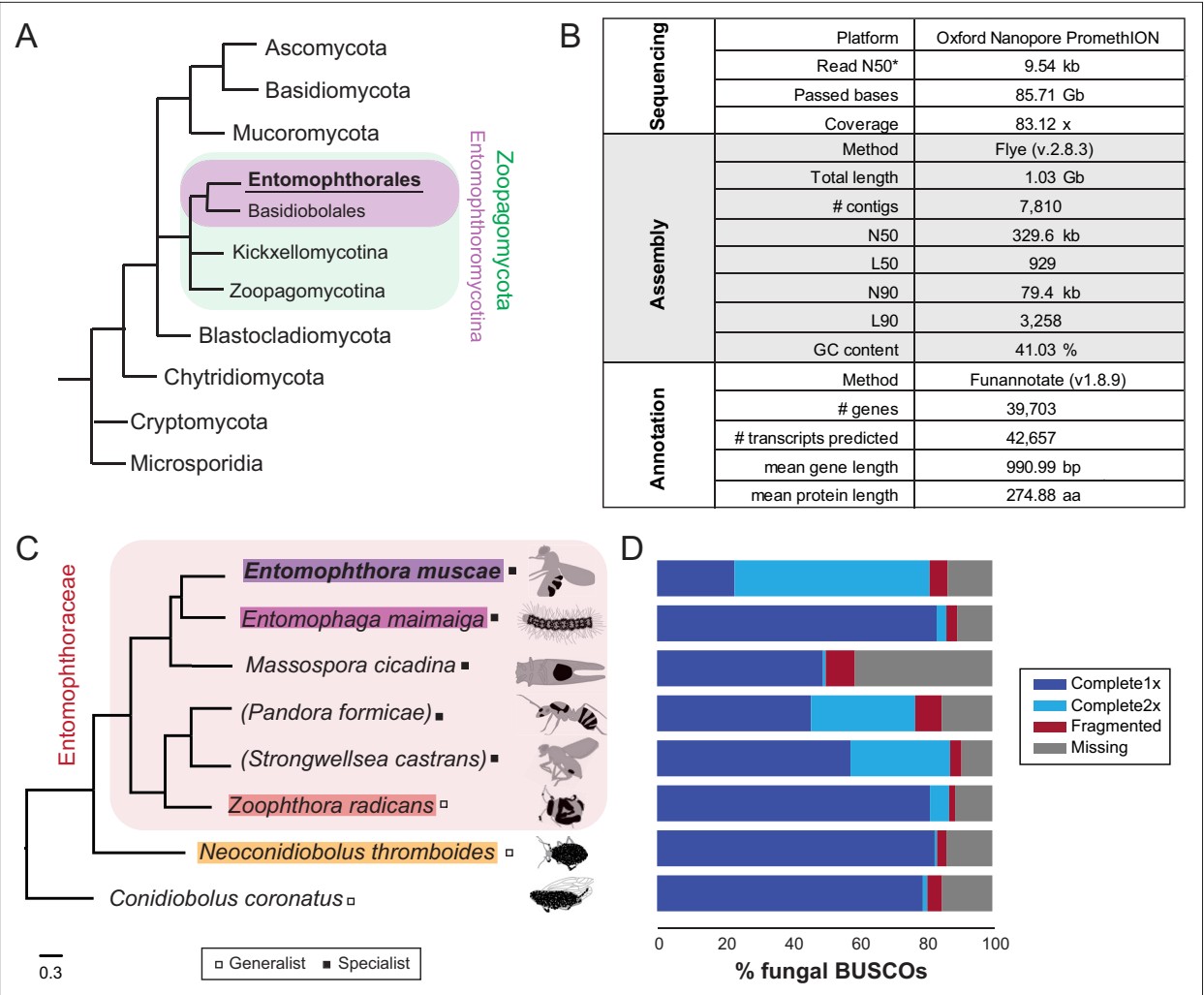

**Figure 1.** A new *Entomophthora muscae* genome assembly and comparative entomophthoralean datasets. (**A**) Fungal cladogram, based on **Spatafora et al., 2016**. All unshaded taxa are fungal phyla. Green-shaded branches are subphyla within phylum Zoopagomycota, purple-shaded branches are orders within subphylum Entomophthoromycotina. Branch lengths are not proportional to phylogenetic distance. (**B**) Overview of sequencing, assembly and annotation statistics for new *E. muscae* genome assembly. Asterisk indicates that this value is for reads that passed the base-calling threshold only, not all bases sequenced. (**C**) Cladogram presenting the evolutionary relationships of the Entomophthorales species considered in this study. Red-shaded branches are genera within the family Entomophthoraceae. Phylogenetic tree was constructed with FastTree using a concatenated set of conserved protein coding genes. Parentheses around *Pandora formicae* and *Strongwellsea castrans* indicate that transcriptomic datasets were used for these species; for all other species genomic datasets were used. Tree branch length is proportional to phylogenetic distance (substitution rate given in legend below). Species whose genomes comprised our core analysis set are colored. Classification of host specificity of each fungus (i.e. specialist or generalist) is denoted with a box to the right of the species name. Specialist species infect a narrow host range; generalists can infect a broad range of species. An example depiction of a host killed by each of these fungi is drawn to the right. Host tissues are gray, with fungal conidiophores depicted in black. Hosts (top to bottom): fruit fly (*Drosophila melanogaster* adult), spongy moth (*Lymantria dispar* larva), periodical cicada (*Magicicada septendecim* adult), ant (*Formica exsecta* adult), cabbage maggot fly (*Delia radicum* adult), Bagrada bug (*Bagrada hilaris* adult), aphid adult (Aphididae), and planthopper (*Delphacodes kuscheli* adult). (**D**) BUSCO completeness estimates for the predicted proteome corresponding to species listed in C, using fungi_odb10 BUSCO set (**Simão et al., 2015**).

The online version of this article includes the following figure supplement(s) for figure 1:

**Figure supplement 1.** Kmer distributions within *E. muscae* genome assembly.

# Results

## A long-read assembly of the E. muscae genome in the context of a comparative Entomophthorales dataset

Using a modified genomic DNA extraction protocol for filamentous fungi in the genus *Trichoderma*

(*Elya and Lee, 2022*), we extracted high molecular weight DNA from a in vitro-grown *Entomophthora muscae* culture inoculated from a single sporulating *E. muscae*-killed fruit fly. Oxford PromethION sequencing of the resultant genomic library yielded 85.71 Gb of sequenced bases (N50=9.54 kb) that passed quality control thresholds (*Figure 1B*). These reads were assembled with Flye (v2.8.3) and self-polished using Medaka (v1.2.6) to yield a 1.03 Gb genome consisting of 7936 contigs (N50=301.1 kb). Additional scaffolding using 10x Genomics linked-read data (SRR18312935; https://ncbi.nlm.nih.gov/sra/?term=SRR18312935) resulted in a final contig count of 7810 (N50=329.6 kb; *Figure 1B*). The assembly was annotated with a custom Funannotate-based pipeline, which included assignment of putative protein functions based on a number of enzyme (e.g. MEROPs *Rawlings et al., 2018*, CAZy *Drula et al., 2022*), protein family (e.g. Pfam *Mistry et al., 2021*, InterPro *Paysan-Lafosse et al., 2023*) and ontology (e.g. GO *Ashburner et al., 2000*; *Aleksander et al., 2023*, EggNog *Huerta-Cepas et al., 2019*, COG *Galperin et al., 2021*; *Tatusov et al., 2000*) databases as well as prediction algorithms for secretion signals (SignalP; *Almagro Armenteros et al., 2019*) and transmembrane domains (see Materials and methods for full details). The final annotation predicted 39,703 genes and 42,657 total transcripts (*Figure 1B*).

To glean biological insights from this new genome, we used a comparative approach to place our new genome in the context of other newly available entomophthoralean data. We collated a dataset consisting of the latest available genomic and transcriptomic data for entomophthoralean fungi, including genomes for the specialist pathogens *E. muscae* (this study, ARSEF 13514), *E. maimaiga* (ARSEF 7190 v1.0) and *Massospora cicadina* (*Stajich et al., 2022*), transcriptomes for the specialists *P. formicae* (*Małagocka et al., 2015*) and *Strongwellsea castrans* (this study) and genomes for the generalists *Zoophthora radicans* (*Amses et al., 2022*), *Neoconidiobolus thromboides* (*Chang et al., 2022*) and *C. coronatus* (NRRL 28638; *Figure 1C*).

To assess completeness of these datasets, their corresponding predicted proteomes were analyzed for the presence of Benchmarking Universal Single-Copy Orthologs (BUSCOs) using a fungal reference set (odb10) (*Manni et al., 2021*). For *E. muscae*, this analysis detected 81.3% of complete fungal BUSCOs and 5.4% fragmented BUSCOs in the *E. muscae* proteome; 13.3% of fungal BUSCOs were not detected (*Figure 1D*). For the other entomophthoralean fungi, percentage of complete-copy BUSCOs ranged from 50.3% (*M. cicadina*) to 87.4% (*S. castrans*), with a median value of 82.4% across this group (*Figure 1D*). Notably only the *E. muscae* genome had more double-copy (58.2%) than single-copy (23.1%) BUSCOs. The next highest proportions of double-copy BUSCOs were found in the transcriptomic datasets of *P. formicae* (31.1%) and *S. castrans* (29.7%) which includes partial transcripts and isoforms in the duplication count. Based on BUSCO scores and phylogenetic positions, we selected three other genomes aside from *E. muscae* to use as a core set for genomic analysis: *E. maimaiga*, *Z. radicans* and *N. thromboides* (*Figure 1C*).

While the high number of duplicate BUSCOs in the transcriptomes could be accounted for by contaminant host transcripts (each of these transcriptomes was assembled from fungus-killed hosts), this explanation could not account for the high number of duplicate BUSCOs in *E. muscae*, which was assembled from the DNA of fungal cells cultured in vitro. One possible explanation for many duplicate BUSCOs in *E. muscae* is that the assembled genome was functionally diploid, not haploid, which has been suggested previously for another *E. muscae* isolate from houseflies (*Musca domestica*; *De Fine Licht et al., 2017*). To address this, we counted the occurrence of kmers (29-mers or 33-mers) throughout our assembly (Jellyfish v.2.3.0 *Marçais and Kingsford, 2011*) and plotted the coverage of kmers against their observed frequency (GenomeScope v2.0 *Ranallo-Benavidez et al., 2020*). We observed a large peak ~50 x coverage, corresponding to single copy (haploid) kmers and a smaller peak at ~100 x coverage, corresponding to dual copy (diploid) kmers. This small ~100 x peak suggests that our assembly is heterozygous at a low level (0.023%, 0.026% respectively) but does not support the hypothesis that our genome is fully diploid (*Figure 1—figure supplement 1A and B*). We also addressed the possibility that differences in how genomes were annotated (i.e. annotation 'pipeline') could lead to inflated gene counts, owing to differences in the stringency of calling genes. In order to address this, we annotated the genomes for *E. maimaga* and *Z. radicans*, which had been annotated with other pipelines, with the pipeline used for *E. muscae*. While our pipeline did predict more genes than the original annotation pipelines for these assemblies (23,807 vs 14,701 for *E. maimaga*; 18,761 vs 14,479 for *Z. radicans*), this difference could not fully account for the discrepancy in gene count (*Supplementary file 1a*). This

suggested that differences in annotation approaches are not sufficient to explain the high duplicate gene count in *E. muscae*.

## Proliferation of transposable elements has led to huge genomes within Entomophthorales

At 1.03 Gb, *E. muscae*'s genome is very large compared to the vast majority of other fungal genomes (*Figure 2A*). Similarly, genomes of the most closely-related entomophthoralean fungi for which genomes are available (*M. cicadina*, *E. maimaiga,* and *Z. radicans*) are also extremely large, each exceeding 500 Mb. However, the impressive sizes of *E. muscae* and other entomophthoralean genomes do not appear to be a consequence of increased gene number (*Figure 2B*). The number of genes predicted in entomophthoralean genomes ranges from 8867 (*N. thromboides*) to 14,701 (*E. maimaiga*) to 39,711 (*E. muscae*). Gene counts for *M. cicadina* are also within the low end of this range (7532), but this number is expected to be an underestimate given the fragmented nature of the assembly. In comparison, fungi with genomes between 10–100 Mb have between ~3000–33,000 genes. Thus, as in other fungi whose genome sizes rank within the top 1% (e.g. the mycorrhizal fungi *Gigaspora margarita* and *G. rosea*, the rusts *Phakospora pachyrhizi*, *Austropuccinia psidii* and *Hemileia vastatrix* and the bioluminescent mushroom *Mycena olivaceomarginata*), the predicted number of genes in entomophthoralean fungi is within the range observed for fungi with more typically sized genomes (<100 Mb).

Unlike gene content, repeat content across entomophthoralean genomes reveals a clear trend: fungi that shared a common ancestor after divergence from *Conidiobolus* have highly repetitive genomes (*Figure 2C*). The genomes of *E. muscae*, *E. maimaiga*, and *M. cicadina* consist of ~90% repeated sequences (90.9%, 90%, and 92.4%, respectively); for *Z. radicans* the proportion of repeated sequences genome-wide is about 20% less, at 71%. Meanwhile, *N. thromboides'* genome is predominantly non-repetitive (only 13.5% sequences are repeated). The bulk of this repetitive content consists of transposable elements, which are categorized as Class I or Class II based on their mechanism of action. Class I transposons (retrotransposons) 'jump' via transcription to an RNA intermediate that is then reverse transcribed to DNA and integrated in a target genomic site. Class II transposons (DNA transposons) can 'jump' by excision from donor DNA and reinsertion into a target genomic site or 'copy' when single-stranded excisions are repaired by host machinery. The majority of repeat content across these four family Entomophthoraceae genomes consists of Class I retrotransposons, specifically Ty3 (formerly called *Gypsy Wei et al., 2022*) long terminal repeat (LTR) retrotransposons. Ty3 elements comprise 39.1%, 43.2%, 56.8%, and 38.0% of the *E. muscae*, *E. maimaiga*, *M. cicadina,* and *Z. radicans* genomes, respectively, while only 0.01% of the *N. thromboides* genome. In addition, the most closely related species pair in this set, *E. muscae* and *E. maimaiga*, also have a sizeable fraction (12% and 13.9%, respectively) of Tc1 mariner DNA (Class II) transposons populating their genomes. *M. cicadina* is unique among the other entomophthoralean fungi in having a substantial fraction (6.3%) of L1 long, interspersed nuclear elements (LINEs) while effectively lacking any DNA transposons.

Kimura divergence estimates for most populous repeat elements of the *E. muscae*, *E. maimaiga*, *M. cicadina*, *Z. radicans,* and *N. thromboides* genomes show trends consistent with phylogenetic relationships between these fungi (*Figure 2D*). All but *N. thromboides* have expanded Ty3 and unknown LTR elements with comparable Kimura divergences (~0–50), indicating similar proliferation of these elements occurred with approximately concurrent timing. *M. cicadina* uniquely shows a burst of Ty3 LTR expansion around divergence time 20 and an expansion of LINE elements, occurring after Ty3 expansion. *E. muscae* and *E. maimaiga* show an increase in TcMar elements more recently than either of the Ty3 and LINE expansions. In addition, *E. muscae*, *E. maimaiga,* and *Z. radicans* share an expansion of DNA-MULE-MuDR elements that is not observed in *M. cicadina*. All five genomes also possess putative repeat elements currently classified as 'Unknown'. These elements appeared roughly concurrently.

Fungi are known to counter the proliferation of TEs by mutating cytosine to thymine in repetitive regions via a process called Repeat Induced Point (RIP) mutations (*Selker, 2002*; *van Wyk et al., 2020*). This process occurs during meiosis in fungi from Ascomycota and Basidiomycota, where homology between DNA regions directs this transition (*Hood et al., 2005*; *van Wyk et al., 2020*). We calculated dinucleotide indices to look for evidence of RIP in entomophthoralean fungi, along with *Neurospora crassa*, an ascomycete fungus in which RIP has been extensively documented (*Freitag*

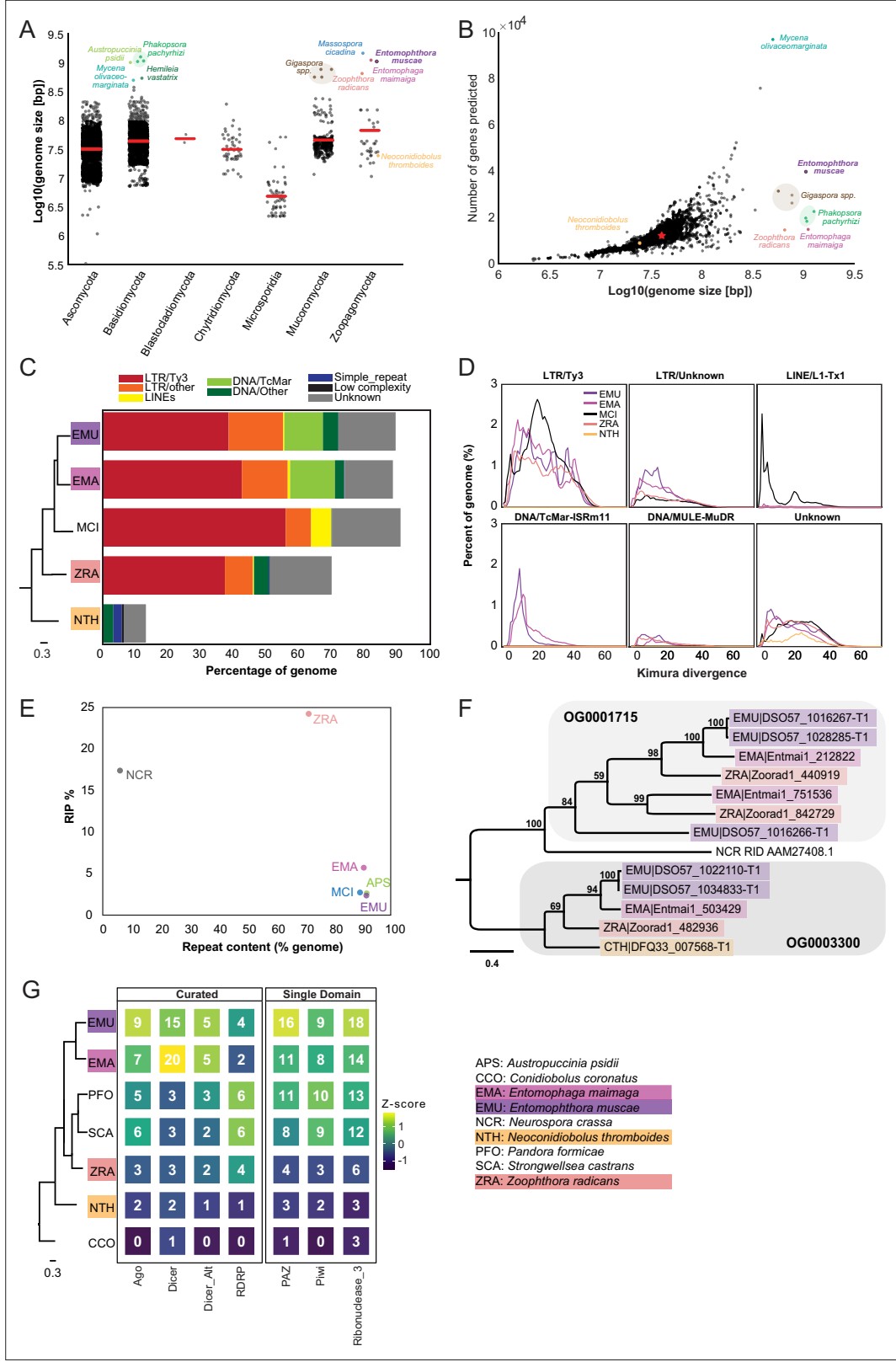

**Figure 2.** Entomophthoralean genomes are enlarged (relative to those of other fungi) due to proliferation of transposable elements. (**A**) Genome assembly sizes of fungal phyla. Red lines indicate mean values per phylum. Species in the core Entomophthorales genomic analysis set are labeled (color-coded per *Figure 1C*). (**B**) Observed genome sizes versus predicted gene number in sequenced fungal genomes. Excluding genomes in excess of

*Figure 2 continued*

500 Mb, the median genome size and number of genes across this dataset is 37.1 Mb and 11,843, respectively (red star). For both A and B, genomes exceeding 500 Mb are indicated by dots colored by species identity. Data for A and B available in **Supplementary file 2**. (**C**) Repeat element composition within *E. muscae* (EMU), *E. maimaiga* (EMA), *M. cicadina* (MCI), *Z. radicans* (ZRA), and *N. thromboides* (CTH) as determined by RepeatMasker (**Smit et al., 2013**). Only repeat elements that exceeded 0.1% of the genome for at least one species are shown. Cladogram modified from **Figure 1C**. (**D**) Landscapes of DNA repeat elements with those comprising less than 1% of DNA repeats in each genome binned as other. (**E**) Percentage of genome in which RIP was detected versus percent of genome comprised of repeat content. (**F**) Protein phylogeny representing relationships among two methyltransferase-containing orthogroups and RID from *Neurospora crassa*: a cytosine methyltransferase required for RIP (**Freitag et al., 2002**). Scale bar indicates 0.3 substitutions per site. (**G**) Counts of selected Pfams associated with RNAi pathway components across genomes. Curated counts include only candidates with the expected combination (and frequency) of Pfams for each of the listed RNAi pathway proteins. Cladogram modified from **Figure 1C**.

---

*et al., 2002*; *Galagan et al., 2003*; *Lewis et al., 2009*; *Selker, 2002*), and *Austropuccinia psidii* (*Tobias et al., 2021*), a basidiomycete rust with a genome of similar size to entomphthoralean fungi (*Figure 2E*). Consistent with previous studies, *N. crassa* showed high levels of RIP (17.4%) (*Cambareri et al., 1991*). Surprisingly, RIP in *Z. radicans* occurred at an even higher rate (24.2%) than in *N. crassa*. We also detected RIP at low levels in the rust *Austropuccinia psidii* (2.6%), *E. muscae* (2.4%), *M. cicadina* (2.7%), and *E. maimaga* (5.7%). This analysis argues that RIP occurs in *Z. radicans* and could be occurring at a low level in other entomophthoralean fungi as well.

A cytosine methyltransferase named RIP defective (RID) is required for RIP (*Freitag et al., 2002*), one of several families of DNA methyltransferases in Fungi (*Bewick et al., 2019*). RID contains a DNA methyltransferase region that is recognized by HMMer as two adjacent Pfam DNA_methylase (PF00145) domains, so we leveraged this characteristic domain profile to identify proteins that share this domain architecture. This revealed two orthogroups (OG0001715 and OG0003300) containing candidates with two DNA_methylase domains. These orthogroups were aligned with RID protein from *Neurospora crassa*, and *N. crassa* RID is sister to the OG0001715 clade in a tree of these proteins, albeit at low (12–14%) sequence identity (*Figure 2F*). OG001715 is composed of proteins from *Z. radicans*, *E. maimaiga* and *E. muscae*. Each of these species has two paralogs in this orthogroup, except *E. muscae*. *E. muscae* intriguingly has an additional, partial paralog, (DSO57_1016266) of about half the size, which is in tandem with another, complete *E. muscae* OG0001715 paralog (DSO57_1016267). This analysis suggests that *E. muscae*, *E. maimaiga*, and *Z. radicans* contain multiple proteins with similar domain architecture as RID, with some direct evidence for duplication of these methyltransferases. We report here on our search for methyltransferases with similar domain signatures to RID, but note that these candidates cannot be interpreted as homologs of RID: their function warrants further investigation.

Many eukaryotic organisms also employ RNA interference (RNAi) to control retrotransposon activity. RNAi occurs when double-stranded RNAs are processed into small interfering RNAs (~20–30 nucleotides long) that are used to guide degradation (or, in the case of mRNA, repression of translation) of complementary RNA. In Fungi, there are three proteins essential for RNAi: (1) RNA-dependent RNA polymerase (RDRP), which processes ssRNA into dsRNA, (2) Dicer, the enzyme that processes dsRNA into siRNA, and (3) Argonaute, a protein that loads siRNA to form an RNA-induced silencing complex (RISC) that identifies target RNAs by complementarity (*Nicolás and Garre, 2016*). We performed a domain-based analysis to assess if RNAi machinery had been lost in entomophthoralean fungi, which could allow transposable elements to proliferate (*Figure 2G*). This analysis suggests that at least one homolog of each of the core components of the fungal RNAi pathway (RDRP, Dicer and Argonaute) is present across our core species, with *E. muscae* and *E. maimaga* predicted to have several homologs of each of these genes. Thus, loss of RNAi pathway genes seems unlikely to account for transposable element proliferation in these species.

## Trends in protein domains across Entomophthorales

We next compared the coding regions of the entomophthoralean datasets in order to identify functional trends across these fungi. Since all of these species are insect pathogens, we expected to observe common themes amongst their metabolic and secreted functions related to their utilization

of host resources and interactions with the host immune system. First, we performed protein domain analysis, looking broadly at protein families (Pfam), and more specifically at functional domains involved in metabolizing carbohydrates (using CAZy) and proteins (using MEROPS). For these analyses we used all available entomophthoralean datasets (*Figure 1C*) except for the *M. cicadina* genome. *M. cicadina* was excluded because the fragmented nature of its genome assembly precluded an accurate prediction of this fungus' proteome, a key requirement for this analysis (*Supplementary file 1b*).

## Protein family domains (Pfam)

Pfam families encompass diverse proteins with a broad range of functions (*El-Gebali et al., 2019*). Enrichment analysis among these domains revealed a unique enrichment in *E. muscae* of retroviral families (i.e. TE-related), specifically Asp_protease (PF09668), Asp_protease_2 (PF13650), gag-asp_protease (PF13975), RVP_2 (PF08284) and dUTPase (PF00692) domains (*Figure 3A*). The domain zf-CCHC (PF00098) was found to be similarly enriched in *E. muscae* and may represent a retroviral regulatory protein. The largest family that was enriched in *E. muscae* was Lipase_3 (PF01764; 348 in *E. muscae*; 5-fold higher than the median across genomes: 66), which was also enriched in *E. maimaiga* and *Z. radicans* (132 and 190, respectively).

A much smaller number of Pfam domains were found to be significantly underrepresented in *E. muscae* (20) compared to the number of domains found to be significantly overrepresented (105) (*Figure 3B*). Proportionally more of these underrepresented families were not predicted to be secreted (17/20, compared to 45/105 of overrepresented Pfam families, p=0.00055 per chi-squared test). Three of these not-secreted, underrepresented families in *E. muscae* were underrepresented by greater than fivefold: alpha/beta hydrolase fold (abhydrolase_3; PF07859; 6.5-fold lower than median), bifunctional feruloyl and acetyl xylan esterases (BD-FAE; PF20434; 20-fold) and endonuclease/exonuclease/phosphatase (Exo_endo_phos_2; PF14529; 13-fold). Alpha/beta hydrolase folds are found as catalytic domains in many different enzymes, including BD-FAE (PF20434) proteins, which specifically act on complex xylans. The BD-FAE family was significantly underrepresented in *E. muscae*, *E. maimaiga*, and *Z. radicans* (1, 2, and 3 BD-FAE-containing proteins, respectively), compared to *C. coronatus* and *N. thromboides* (23 and 25, respectively). *E. muscae*, with only a single predicted BD-FAE-containing protein, was additionally significantly lower than *P. formicae* and *S. castrans*, which had 20 and 23, respectively.

We compared the broad Pfam composition of these seven genomes (*Figure 3C*). A total of 2684 Pfam domains were present in all genomes. *S. castrans* and *P. formicae* shared a unique overlap of 1,142 domains, each with 170 and 485 unique domains, respectively. *E. muscae*, by comparison, only had 14 unique Pfam domains. The trend of *P. formicae* and *S. castrans* sharing many domains was seen across all domain types. However, the proteomes for these fungi were uniquely generated from transcriptomic data, so we were concerned this pattern may reflect a methodological, not biological, signal. To test this, we recapitulated our Pfam domain analysis incorporating a proteome generated from transcriptomic data collected from *E. muscae*. The results of this analysis are summarized in an UpSet plot (*Figure 3—figure supplement 1*), in which the *E. muscae* transcriptomic dataset does not meaningfully group with *P. formicae* and *S. castrans*. This suggests that the unique subset shared between *P. formicae* and *S. castrans* is likely due to their evolutionary history, not a methodological artifact.

## Carbohydrate metabolic domains (CAZy)

The CAZy data set (named for Carbohydrate-Active Enzymes) provides insight into which carbohydrates can be metabolized. The majority (55) of CAZy domains were shared among all fungi, with *S. castrans* and *P. formicae* sharing a large set of CAZy enzymes (28) absent in the other species. This unique pattern for *S. castrans* and *P. formicae* was consistent across all types of domains and shows a clear signal on the UpSet plot comparing domains (*Figure 3C*). Six CAZy domains were found to be significantly enriched in *E. muscae*, including AA11 (lytic chitin monooxygenases), AA7 (glucooligosaccharide oxidases), CBM18 (chitin-binding and chitinases), CE16 (acetylesterases), CE4 (acetyl xylan esterases), and GT1 (UDP-glucuronosyltransferase; *Figure 3—figure supplement 2A*). AA11 and CE16 were notably significantly enriched in *E. muscae*, *E. maimaiga* and *Z. radicans* and significantly lower in *P. formicae* and *S. castrans*.

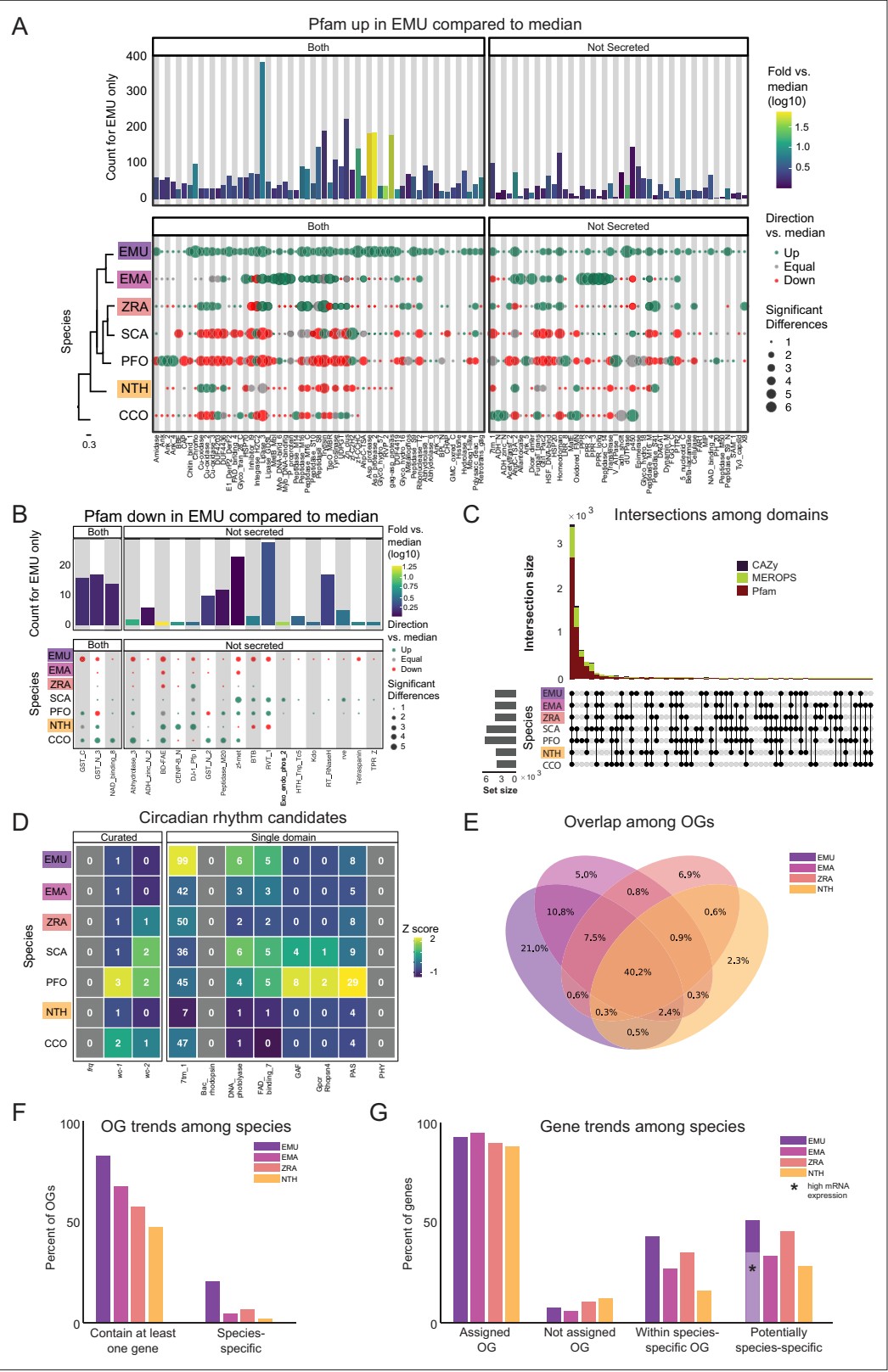

**Figure 3.** Comparison of domain architecture/gene content across Entomophthorales. (**A**) Pfams significantly overrepresented in *E. muscae* (EMU) compared to other species analyzed *E. maimaiga* (EMA), *Z. radicans* (ZRA), *S. castrans* (SCA), *P. formicae* (PFO), *N. thromboides* (NTH), and *C. coronatus* (CCO). Bars represent the counts for *E. muscae* colored by fold-versus-the-median across all genomes. Point size represents the number

*Figure 3 continued on next page*

*Figure 3 continued*

of significant pairwise comparisons among other genomes and these are colored according to whether the value is above, below or equal to the median value across all genomes. Cladogram modified from *Figure 1C*. (**B**) Pfams significantly underrepresented in *E. muscae* compared to other species analyzed. Plot format as in A. (**C**) Combined UpSet plots showing the intersection among genomes for all domain categories (CAZy, Pfam and MEROPS; bar colors). (**D**) Counts of selected Pfams associated with circadian proteins across genomes. Curated counts include only candidates with the expected combination of Pfam domains for each of the listed circadian proteins. (For example, to be considered a curated *wc-1* candidate, a gene needed to have one each of GATA, PAS_3 and PAS_9 domains.) (**E**) Venn diagram depicting intersections between predicted OGs among *E. muscae*, *E. maimaiga*, *Z. radicans,* and *N. thromboides*. Values only within a single ellipse indicate the abundance of species-specific genes. (**F**) Occupancy trends by species for 11,7111 OGs recovered from amalgamating *E. muscae*, *Entomophaga maimaiga*, *Z. radicans* and *N. thromboides* gene models. 'Contain at least one gene': percentage of OGs that contain at least one gene from a given species. 'Species-specific' is the percentage of OGs that only contain genes from one species. (**G**) Percentage of genes for each species found in each OG type as percentages of total genes annotated in the genome. 'Assigned OG' are genes that clustered with anOG; 'Not assigned OG' are genes that did not cluster with any OG. 'Within species-specific OG' reflects the percentage of genes that fall within an OG that is only populated by genes of the given species. 'Potentially species-specific' is the sum of genes 'Not assigned OG' and 'Within species-specific OG'. The light purple bar marked with a black asterisk indicates the percentage of genes that are potentially species-specific with evidence of high expression in an in vivo dataset (expression <5, pooled dataset of 27 whole fly samples exposed with *E. muscae*).

The online version of this article includes the following figure supplement(s) for figure 3:

**Figure supplement 1.** Pfam UpSet plot analysis including *E. muscae* transcriptome.

**Figure supplement 2.** Additional domain analysis for CAZy and MEROPS databases.

**Figure supplement 3.** *E. muscae* core OG Pfam enrichment.

## Peptidase domains (MEROPS)

All genomes included in our domain analysis shared 691 MEROPS peptidase domains (*Figure 3C*). *S. castrans* and *P. formicae* shared 448 MEROPS domains absent in the other genomes, with each having a large number of unique MEROPS domains (173 and 136, respectively). *E. muscae* by contrast had about a third to a quarter as many unique MEROPS peptidases (37). Among *E. muscae* MEROPS, 92 were found to be significantly different in comparisons among genomes, with 74 MEROPS domains significantly enriched and 9 domains having significantly fewer proteins (*Figure 3—figure supplement 2B*). A clear trend for *E. muscae*, *E. maimaiga* and *Z. radicans* is enrichment in M16A metallopeptidases (i.e. MER0001214, MER0001218, MER0002283, MER0002345, MER0002423, MER0003386, MER0003823, MER0011096, MER0011744, MER0015259, MER0169735) compared to *S. castrans*, *P. formicae,* and *N. thromboides*. Considering proteins containing these 10 M16A domains, *E. muscae* (62 on average), *E. maimaiga* (19), and *Z. radicans* (44) were comprised of approximately 11-fold more proteins (43 vs 4) compared to *S. castrans* (5), *P. formicae* (3) and *N. thromboides* (2). Two metallopeptidases were uniquely enriched in *E. muscae*: MER0001120, a pro-collagen C-peptidase, and MER0016735, an M23 peptidase. Among serine peptidases, a lipid monooxygenase (MER0031618), two S33 prolinases (MER0036050 and MER0036051) and three kexin/kexin-like (MER0000364, MER0000374 and MER0001604/krp1) domains were found to be significantly enriched. Additionally, *E. muscae* was uniquely enriched in the monoglyceride lipase MER0045883. Considering cysteine peptidases, *E. muscae* had a high number of metacaspases across all 10 families, though fewer than the number seen in *E. maimaiga* (~50 accessions for each family). Only a handful of domains were found to be significantly underrepresented in *E. muscae*, including aminopeptidase H11 (MER0002003), cystinyl aminopeptidases (MER0002060), and ERAP2 aminopeptidases (MER0002968) (*Figure 3—figure supplement 2C*).

## Evidence for circadian pathways

We also examined genes that may be involved in the ability to sense environmental light cues and maintain circadian time, as these processes are likely involved in fungal manipulation of host behavior (*de Bekker et al., 2021*; *de Bekker and Das, 2022*). The circadian clock of *Neurospora crassa* (Ascomycota) is the most highly studied and best understood circadian clock among Fungi (see *Cha et al., 2015* for overview). In *N. crassa*, two photosensitive proteins, White Collar 1 and 2 (WC-1 and WC-2)

form a heterodimeric complex (called the White Collar Complex, WCC). In the presence of light, WCC undergoes a conformational change and becomes a transcriptional activator of light-regulated genes, including the gene *frequency* (*frq*) that encodes an intrinsically-disordered protein. FRQ protein forms a heterodimeric complex with its partner, FRQ-interacting RNA Helicase (FRH), and this complex, called FFC (FRQ-FRH Complex), inhibits the activity of WCC. After protein synthesis, FRQ becomes progressively more phosphorylated, eventually leading to its degradation and reduced concentration of FRQ and FFC. When FRQ titers drop below a certain threshold, WCC activity can proceed unimpeded, thus closing an oscillating cycle of transcription and translation with a periodicity of about 24 hr.

To identify potential components of circadian oscillators in our entomophthoralean datasets, we checked for putative homologs of *wc-1*, *wc-2 and* frq and domains commonly found in circadian proteins, per-ant-sim or PAS (PAS_3: PF08447) and the structurally similar domain GAF (GAF: PF01590; *Figure 3D*). All the analyzed entomophthoralean datasets had at least one copy each of a gene encoding for a protein with the same domains found in *N. crassa* WC-1, with *C. coronatus* and *P. formicae* containing multiple genes (the latter even having two candidate WC-1 genes in tandem, Pfor_14152, Pfor_14153). For *wc-2*, all but *E. muscae*, *Entomophaga maimaiga* and *N. thromboides* also had at least one candidate gene. None of these species contained a gene encoding for a protein with the hallmark domains of *frq* (*Figure 3D*).

In addition to finding evidence for at least one copy of a *wc-1* homolog for all of the species analyzed, we also searched for genes encoding for known light-sensitive domains (rhodopsin family [7tm_1; PF00001], bacteriorhodopsin [Bac_rhodopsin; PF01036], DNA photolyase [DNA_photolyase; PF00875], the FAD-binding domain of DNA photolyase [FAD_binding_7; PF03441] and phytochrome [PHY; PF00360], GPCR rhodopsin 4 [GpcrRhopsn4; PF10192]). We did not detect any bacteriorhodopsins or phytochromes within our datasets, but we found several instances of genes containing the other light-sensitive domains, with *E. muscae* having far more than the rest (99 genes with rhodopsin family domains as well as 11 photolyases or FAD-binding domains thereof; ) (*Figure 3D*). This analysis suggests that each of these fungi detect light. As light is the predominant cue for setting the phase and period of circadian clocks, this is consistent with the notion that these fungi may also be able to keep time.

## Orthogroup analysis reveals core functionalities across Entomopthorales fungi

We next assessed gene homology among the core genomes in our comparative dataset: *E. muscae*, *E. maimaga*, *Z. radicans,* and *N. thromboides* (*Supplementary file 1b*). For this analysis, we used OrthoFinder to identify orthogroups (sets of genes that are descended from a single gene in the last common ancestor of all species within our analysis set, including both orthologs and paralogs, referred to henceforth as OGs) among the pooled genes from this set of genomes (*Emms and Kelly, 2015*). We reasoned that such an analysis could be a strategy to identify core genes important for an obligate entomopathogenic lifestyle, and conversely, could reveal potentially unique genes in each of the species examined, specifically in our focal species *E. muscae*. We recovered 17,111 OGs. A total of 6878 OGs (40.2%) contained representatives in all four species (*Figure 3E*), and we refer to these as 'core' OGs.

To gain some insight into the function of the core OG genes in *E. muscae,* we performed a Pfam annotation enrichment analysis. Twelve Pfam domains were significantly overrepresented in these genes (*Figure 3—figure supplement 3*) by factors ranging from 1.52 to 2.35-fold. These domains are found in proteins across a variety of principal cellular functions, spanning DNA replication, transcription, macromolecular assembly and signal transduction and included kinases (PF0069, PF07714), RNA recognition and binding proteins (PF00076), helicases (PF00400, PF00271, PF04851), endonucleases (PF04851), ATPases (PF00004), Beta-propeller (PF00400), phosphatidyl inositol-binding (PF00169), RNA splicing (PF00176), and DNA replication (PF00226). Genes that lacked a Pfam assignment altogether were underrepresented (odds-ratio 0.46), consistent with core OG genes belonging to conserved cellular processes among these four species. Sixteen Pfam domains were significantly underrepresented, ranging from being completely absent to 18.3–1.83-fold underrepresented (odds-ratio 0.055–0.55, respectively). Protein domains that were completely absent included chitin recognition (PF00187), transferase (PF02458), multicopper oxidases (PF00394 and PF07732) and polysaccharide

deacetylase (PF01522). Underrepresented domains tended to be related to metabolism and transcription factors (PF00046 and PF00172). However, underrepresented genes also included the retrotransposon gag protein, which is involved in activity of retrotransposons (such as LTRs), and rhodopsin family genes.

The next largest overlap consisted of 10.8% of OGs that are shared only between the most closely related species in this set, *E. muscae* and *Entomophaga maimaiga* (*Figure 3E*). OGs shared between *E. muscae*, *Entomophaga maimaiga* and *Z. radicans*, but not the relative outlier *N. thromboides* totaled 7.5%, with the next largest overlapping set consisting of 2.4% of OGs that are shared between *E. muscae*, *Entomophaga maimaiga* and *N. thromboides*, but not with *Z. radicans*. Other overlapping sets individually accounted for no more than 0.9% of all OGs. The highest proportion of species-specific OGs (OGs that are populated by genes of a single species) was found for *E. muscae* (21%), followed by *Z. radicans* (6.9%), *Entomophaga maimaiga* (5%), and *N. thromboides* (2.3%). Accordingly, we observed that *E. muscae* genes also populate the most OGs overall, although has genes in multi-species OGs no more frequently than *Entomophaga maimaiga* (*Figure 3F*). These observations are consistent with the *E. muscae* genome containing about three times as many annotated genes as any other fungus considered.

Across these four species, 88–94.4% of genes were assigned to OGs (*Figure 3G*). With respect to species-specific OGs, *E. muscae* has the most: 43.5% of *E. muscae* genes are found in these OGs. The species with the least, *N. thromboides*, had just 16.3%. For each species, the genes that either failed to cluster with an OG (were not assigned) together with the genes that are found in species-specific OGs comprise a set of genes that are potentially unique to that species. If we consider all of the annotated genes in the *E. muscae* assembly, *E. muscae* has the highest percentage of species-specific genes among this species set at 50.9% (*Figure 3G*). However, if we filter our annotated genes using a pooled in vivo transcriptomics dataset (*Elya et al., 2018*), we find that the percentage of species-specific genes for *E. muscae* falls to 35%, within the range of the other fungal genomes analyzed.

## Gene expansions and specializedmetabolite production in *E. muscae* and Entomophthorales

Because *E. muscae* is an obligate insect-pathogen (e.g. can only complete its life cycle within live flies), we investigated the presence of canonical entomopathogenic enzymes in the genome. We found that *E. muscae* appear to have an expanded group of acid-trehalases compared to other entomopathogenic and non-entomopathogenic Entomophthorales (*Figure 4A*), which correlates with the primary sugar in insect blood (hemolymph) being trehalose (*Thompson, 2003*). The obligate insect-pathogenic lifestyle is also evident when comparing the repertoire of lipases, subtilisin-like serine proteases, trypsins, and chitinases in our focal species versus Zoopagomycota and Ascomycota fungi that are not obligate insect pathogens (*Figure 4B*). Sordariomycetes within Ascomycota contains the other major transition to insect-pathogenicity within the kingdom Fungi (*Araújo and Hughes, 2016*). Based on our comparison of gene numbers, Entomophthorales possess more enzymes suitable for cuticle penetration than Sordariomycetes (*Figure 4B*). In contrast, insect-pathogenic fungi within Hypocreales possess a more diverse secondary metabolite biosynthesis machinery as evidenced by the absence of polyketide synthase (PKS) and indole pathways in Entomophthorales (*Figure 4C*).

## Features of *E. muscae* potentially distinct among Entomophthorales

To investigate unique characteristics of the *E. muscae* proteome, we identified all domains that were either unique to (*Figure 5A*) or missing from (*Figure 5B*, Fig. S5-1) *E. muscae*. Among the 41 domains unique to *E. muscae* (*Figure 5A*), the Pfam family Peptidase_A2B (PF12384), a component of Ty3 retrovirus-like elements, was by far the most common (26 proteins). Many Pfam families that were uniquely found in *E. muscae* were apparently related to retrotransposons. Thioredoxin_4 (PF13462) was the second-most abundant unique Pfam family (6 proteins). Among other *E. muscae*-specific domain families, two proteins were annotated as containing abhydrolase_8 domains (PF06259), perhaps accommodating for the apparent significant reduction in abhydrolase_3 domains in *E. muscae* (*Figure 3B*). Nearly all MEROPS families that were unique to *E. muscae* were serine peptidases. A single metallopeptidase (MER0001219; M16A family Axl1 peptidases), threonine peptidase (MER0026205; T3 family) and cysteine peptidase (MER0014097; C19 family) were found uniquely in *E. muscae*. The most abundant MEROPS family that was unique to *E. muscae* was a chymotrypsin family

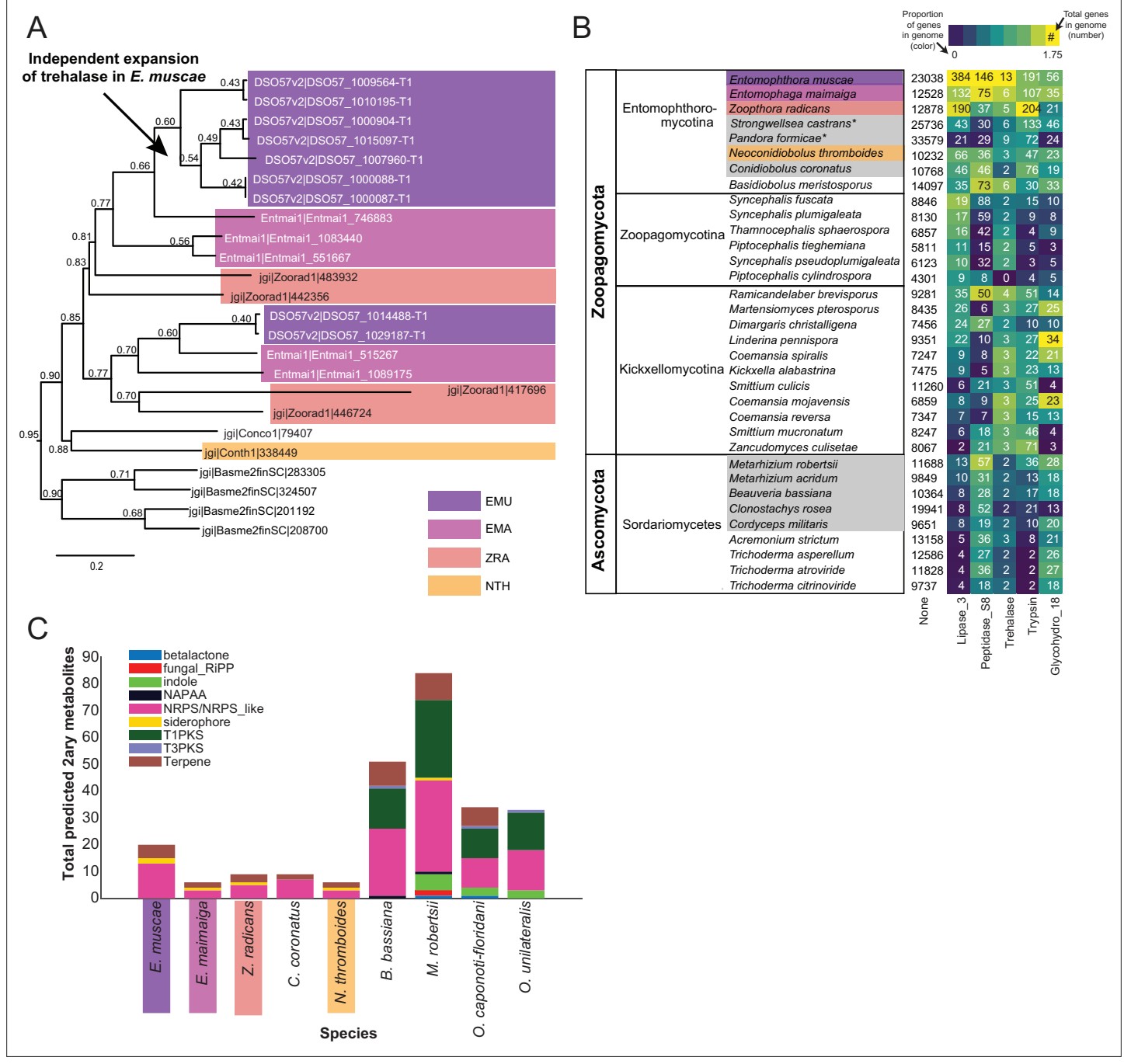

**Figure 4.** Gene family expansion and secondary metabolite production of *E. muscae* and other insect pathogens. (**A**) A family of genes encoding extracellular trehalase enzymes (PF01204) is expanded in *E. muscae* (EMU) compared to other Zoopagomycetes: *E. maimaiga* (EMA), *Z. radicans* (ZRA), *N. thromboides* (NTH), *C. conidiobolus* (Conco1), and *B. meristosporus* (Basme2finSC). (**B**) Total number of genes and number of genes encoding Lipases (*Lipase_3*), Subtilisin-like serine peptidases (*Peptidase_S8*), Trehalases (*Trehalase*), Trypsins (Trypsin), and Chitinases (*Glycohydro_18*) in representative fungal species of Zoopagomycota and Ascomycota. Fungal species in gray are insect pathogens and the four Entomophthoromycotina species are outlined in the same colors as A. Numbers inside heatmap refer to the number of genes that encode a given Pfam domain, and color scale refers to proportion of genes with a given Pfam compared to the total number of genes in the genome (in percentages). (**C**) Predicted secondary metabolite production for select entomophthoralean genomes (*E. muscae*, *E. maimaiga*, *Z. radicans*, *C. coronatus* and *N. thromboidies*) and common ascomycete entomopathogens (*B. bassiana*, *M. robertsii*, *O. caponoti-floridani*, *O. unilateralis*), as predicted by AntiSMASH. Color indicates metabolite class.

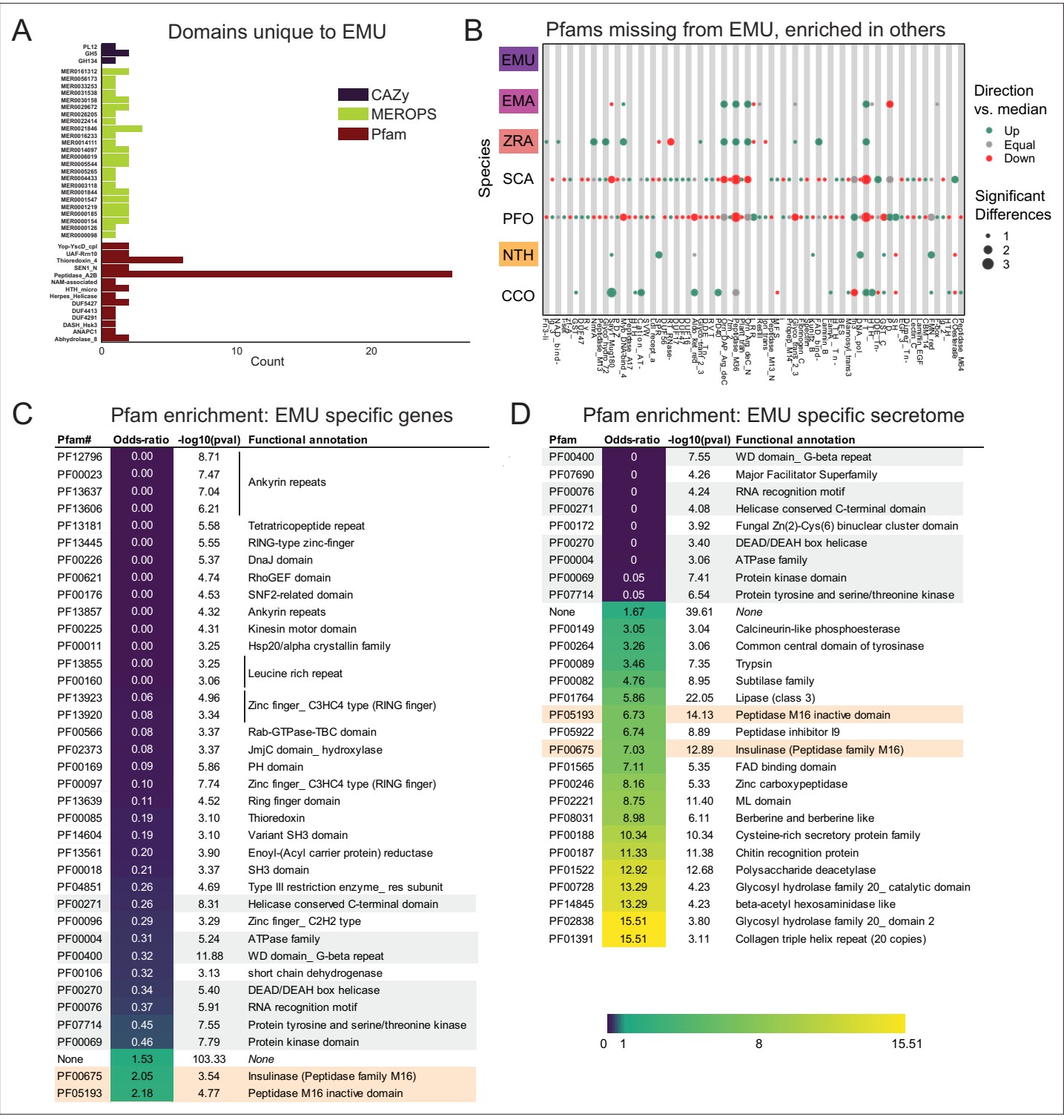

**Figure 5.** Unique features of *E. muscae* compared to *E. maimaiga*, *Z. radicans* and *N. thromboides*. (**A**) Domains unique to *E. muscae*. (**B**) Pfam domains that are missing in *E. muscae*, but enriched in other entomophthoralean fungi. (**C**) Significantly enriched Pfam domains (p ≤ 0.001) within genes that are potentially *E. muscae*-specific (both genes that did not cluster with any orthogroup and genes that cluster with orthogroups that are species-specific; N=9,150 genes). (**D**) Significantly enriched Pfam domains (p-value ≤ 0.001) within potentially *E. muscae*-unique genes encoding proteins predicted to be secreted (N=1685 genes). Odds-ratios are colored according to the scale bar to bottom right. Two Pfam domains (PF00675 and PF05193; highlighted orange in C and D) are overrepresented in potentially *E. muscae*-specific *E. muscae* genes both genome-wide and within the predicted secretome. Pfam domains highlighted in gray are underrepresented across both of these sets.

*Figure 5 continued on next page*

*Figure 5 continued*
The online version of this article includes the following figure supplement(s) for figure 5:
**Figure supplement 1.** CAZy and MEROPs domains missing in *E. muscae* and enriched in other fungi.

(3 proteins; MER0021846). Only three CAZy domains were uniquely found from *E. muscae*: GH134 (containing endo-β–1,4-mannanases), GH5 (glucanases/cellulases) and PL12 (heparin-sulfate lyase).

In total, 2237 Pfam domains were absent in the *E. muscae* genome but observed in other genomes within our core set of entomophthoralean fungi. To assess loss of key domains in *E. muscae*, we considered domains that were missing from *E. muscae*, but were significantly enriched in other entomophthoralean fungi (*Figure 5B*, *Figure 5—figure supplement 1*). We did not observe any domains across Pfam, MEROPS and CAZy that were both missing from *E. muscae* and overrepresented across all other entomophthoralean fungi, suggesting that *E. muscae* has not lost any functions specific to the Entomophthorales. Our domain analysis suggests that *E. muscae* is unique largely through its protein-domain family expansions, not in its losses of domains. Many domains, of all types in our analysis, that were not enriched (or not present) in *E. muscae* were enriched only in *P. formicae* and *S. castrans*, suggesting this signal is reflective of functionalities unique to those species, rather than functionalities lost in *E. muscae*.

We also performed an enrichment analysis on the orthogroups that were classified as potentially species-specific for *E. muscae*. As in our previous OG analysis, we considered genes to be potentially species-specific if they either failed to cluster with an OG in *E. maimaiga*, *Z. radicans* or *N. thromboides* or clustered with OGs comprising only other *E. muscae* genes. We performed an enrichment analysis of Pfam annotations of genes that met these criteria, comparing the frequency of Pfam occurrence in this set against occurrence within all annotated genes (*Figure 5C*). We observed significant overrepresentation of three Pfams among potentially *E. muscae*-specific genes: insulinases (PF00675) M16 peptidases (PF05193) and genes that lacked any Pfam annotation. That *E. muscae* may have specific peptidases is consistent with our understanding of the entomopathogenic lifestyle: these enzyme families are both key in host recognition and invasion as well as host resource utilization (*Arnesen et al., 2018*). Finding genes lacking any Pfam annotation as another enriched group is consistent with the large number of genes observed in *E. muscae* compared to other species, and suggests that there are novel genes of undescribed/unknown function specific to *E. muscae*.

Pfams that were underrepresented in the set of potentially *E. muscae*-specific genes spanned multiple processes, but a few general themes emerged. These include a lack of proteins with cytoskeletal (e.g. ankyrin [PF12796, PF00023, PF13637, PF13606, PF13857], kinesin [PF00225]), cell signaling (e.g. RhoGEF [PF00621], Rab-GTPase [PF00566], proteins tyrosinase and kinase [PF07714, PF00069], thioredoxin [PF00085], pleckstrin homology domain [PF00169]), and transcriptional regulation functions (e.g. zinc fingers [PF13445, PF13923, PF13920, PF00097, PF13639, PF00096], SNF2-related [PF00176], DEAD box helicase [PF00270], RNA recognition motif [PF00076], JmjC [PF02373]). The underrepresentation of these domains in potentially *E. muscae*-specific genes suggests their functionalities are similar to those in other entomophthoralean species.

We performed a similar enrichment analysis looking at the set of potentially *E. muscae*-unique genes that are predicted to encode secreted proteins (*Figure 5D*). We observed several Pfams that were underrepresented within the predicted secretome, many which encompassed functions that are not expected to serve the fungus in an extracellular context (e.g. domains involved in gene transcription [PF00076, PF00172, PF00270] and ATP production [PF00004]). We also observed a few underrepresented Pfam domains involved in signaling ([PF00069, PF07714]). Pfams could be underrepresented in *E. muscae*-specific secreted proteins either because similar proteins are present across other entomophthoralean fungi considered or because proteins containing these domains are not typically secreted. We suspect the Pfams involved in signaling are underrepresented due to their similarity to proteins in other entomophthoralean fungi, while Pfams involved in intrinsic (i.e. not extracellular) processes are underrepresented due to a both unlikelihood of being secreted and conserved function among species.

Pfams that were significantly overrepresented in the predicted *E. muscae*-unique secretome included various catabolic enzymes (e.g. phosphoesterase [PF00149], tyrosinase [PF00264], proteases including trypsin [PF00089], subtilase [PF00082], peptidase inhibitor I9 (known constituents of subtilisins) [PF05922], zinc carboxypeptidase [PF00246], insulinase [PF00675], lipase [PF01764],

polysaccharide deacetylase [PF01522], glycosyl hydrolases [PF00728, PF02838]) and macromolecular recognition domains (e.g. chitin recognition, ML domain). The observation that catabolic enzymes are overrepresented within the putatively *E. muscae*-specific predicted secretome is consistent with our current understanding of entomopathogen host interactions (*Elya and De Fine Licht, 2021*). These enzymes are important during fungal infection, where they help degrade chitin-linked proteins in the cuticle when the fungi force their way inside insects. Both during growth on the cuticle and when proliferating inside the living insects *E. muscae* use enzymes such as peptidases, glycosyl hydrolases etc. to obtain nutrients that support fungal growth in the hemocoel (*Elya and De Fine Licht, 2021*). In addition, we saw enrichment of the collagen triple helix repeat domain (PF01391), FAD binding domain (PF01565) and berberine-like proteins (PF08031). As with our analysis of all *E. muscae*-unique genes, an overrepresentation of genes that were not associated with any Pfam suggests that many *E. muscae* genes have as yet unknown functions.

## Morphological characters and sequence data suggest divergent phylogenetic relationships for *E. muscae* species complex strains

To compare the performance of morphology-based identification with DNA sequence, we mined NCBI GenBank for sequence data for members of the *E. muscae* species complex (EMSC) and outgroup taxa within the genus *Entomophthora*. Novel Sanger sequence data was also generated in support of this analysis, which has been deposited in NCBI (i.e. ARSEF_13514, ARSEF_6918, SoCal_c1 and LTE_c1, See Materials and methods). In parallel, BLASTn searches using reference sequences generated from *E. muscae* ARSEF 13514 were used to identify closely related sequence matches for both ITS and 28 S. Sequences were aligned to assess length and quality, and metadata was compiled to determine which strains had available sequence data and morphological data. Fourteen strains were chosen for inclusion based on the availability of both DNA sequence data and morphological data.

Two methods of phylogenetic inference, maximum likelihood (ML) and Bayesian inference (BI), for the 2-gene concatenated dataset resolved the EMSC as a well-supported monophyletic group that includes *E. ferdinandii, E. muscae,* and *E. scatophagae* (*Figure 6A*). In addition, *E. schizophorae* and the clade containing *E. syrphi* and *E.* aff. *grandis* were both monophyletic and well-supported by both phylogenetic methods. Within the EMSC, two well-supported clades were observed, referred to hereafter as EMSC Clade 1 and EMSC Clade 2 (*Figure 6A*). Clade 1 contained ARSEF 13514 from *D. melanogaster* (Drosophilidae) and three other strains from Muscidae (*Figure 6B*). Clade 2 had host species from three fly families (Anthomyiidae, Muscidae, and Scathophagidae) and contained strains from *E. ferdinandii* and *E. scatophagae* in addition to strains identified as *E. muscae*. Strains sequenced for this study from various *Drosophila* spp. also expanded the known ranges of *E. schizophorae* and *E. grandis* (*Keller, 2002*).

We examined the number of nuclei in, and dimensions of, primary conidia, which are potentially diagnostic morphological characters for these species (*Keller, 2002*). For the six formally described fly-infecting *Entomophthora* spp. included in the study, there was considerable overlap in the ranges of size and mean number of nuclei from primary conidia (*Figure 6B and D*). The ranges of at least four species overlap among the included *Entomophthora* species. The published sources of data used for this analysis varied in their measures (mean, maximum, minimum, standard deviation, and standard error) for number of nuclei and sample sizes. Nuclei data was missing for 3 of 14 strains. Number of nuclei was the most commonly reported morphological character, but this measure alone did not resolve species boundaries, except for *E. schizophorae*.

Primary conidial length and width measurements provided better species-level resolution, but these measurements were only available for 4 of 14 strains (*Figure 6C*). Reported spore measurement ranges for known species overlaid with those included in this study are not aligned with phylogenetic findings. Strains ARSEF 13514 and ARSEF 6918, which occurred in EMSC Clade 1 and EMSC Clade 2, respectively, overlapped with measurements for *E. ferdinandii*, but not *E. muscae*. Strains ARSEF 6716 and KVL14-17, both members of EMSC Clade 1, had overlapping spore measurements with each other and *E. muscae,* but not *E. ferdinandii*. These spore measurements further support *E. scatophagae* as part of the larger EMSC. They also highlight significant overlap between *E. grandis* and *E. syrphi*.

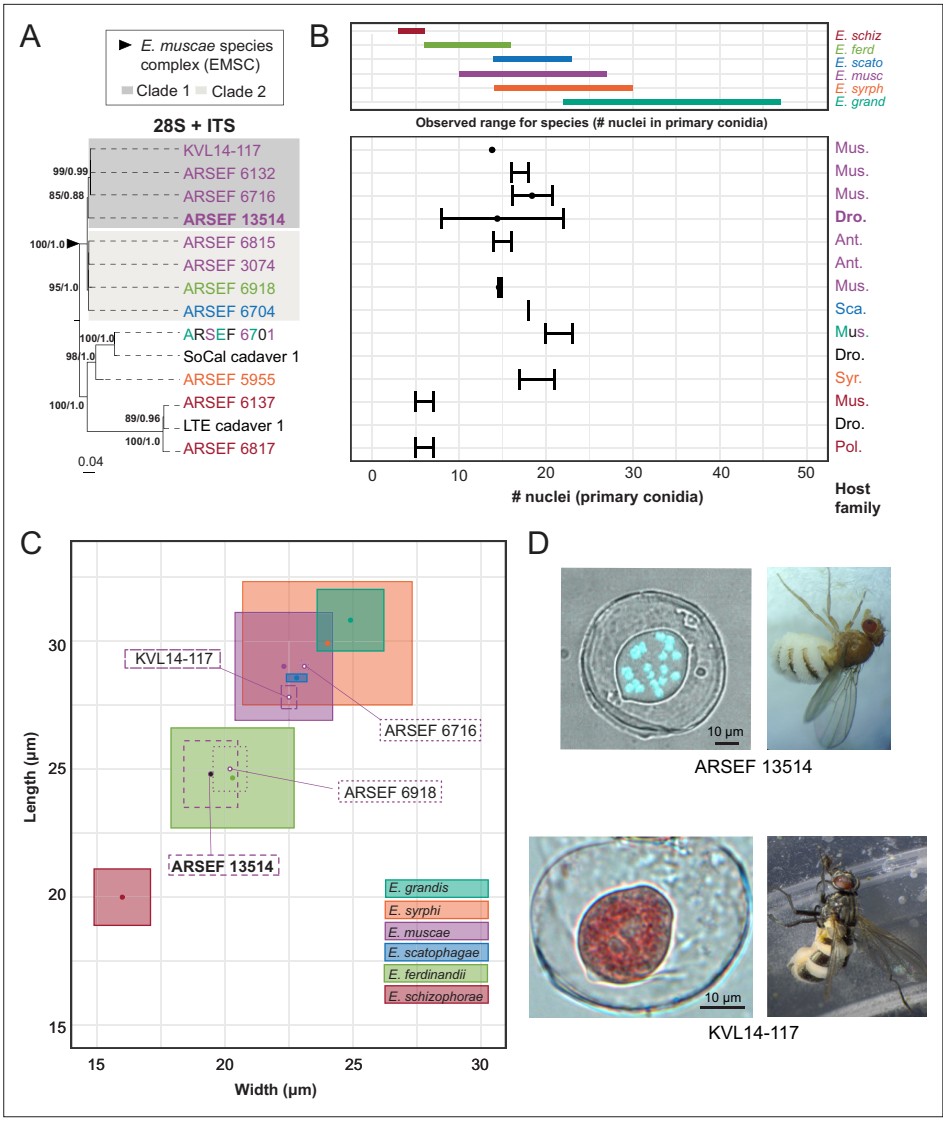

**Figure 6.** Phylogenetic and morphological data for *E. muscae* ARSEF 13514, members of the *Entomophthora muscae* species complex, and closely allied fly-infecting *Entomophthora* spp. Across all panels, species are designated by color (see key in B). (**A**) Concatenated ITS +28 S phylogenetic tree of representative *Entomophthora* spp. including diverse strains across the EMSC. Gray boxes indicate distinct well-supported clades within the EMSC. Topology and branch lengths shown are from the ML analysis. Bootstrap support and posterior probabilities are indicated near each node (ML / BI), and only nodes with >50% support are labeled. ARSEF 6701 is denoted by three colors to indicate that this strain has multiple identifications (*E. grandis* (teal), *E. muscae* (purple) and *E.* sp. (black)). (**B**) Nuclei number of primary conidia among strains (bottom) relative to the known ranges for each of six described species included in this study as defined by **Keller, 2002** (top). Fly family is noted at the far right for each strain (Mus.=Muscidae, Dro.=Drosophilidae, Ant.=Anthomyiidae, Sca.=Scathophagidae, Syr.=Syrphidae, and Pol.=Polleniidae). For panel A, ITS and 28 S sequence data for ARSEF 13514 (ITS and 28 S), ARSEF 6918, SoCal cadaver 1 and LTE cadaver 1 are original to this study, while data for the remainder are from the literature (see **Supplementary file 1c**). (**C**) Primary conidial length and width for strains with published spore measurements, overlaid atop measurements reported for each of the six species. Isolate in bold is *E. muscae* described in this study. (**D**) Primary conidia (left) and fly hosts (right) for *E. muscae* strains ARSEF 13514 (above) and KVL14-117 (below). The *E. muscae* ARSEF 13514 primary conidium is stained with Hoechst 33342 to visualize multiple nuclei contained within conidium. The *E. muscae* KVL 14–117 conidium is stained with aceto-orcein. Scale bars: 10 µm.

## Discussion

### A new entomophthoralean genome

We report a new ~1 Gb assembly of the *E. muscae* genome obtained using a long-read sequencing approach. Despite the comparative ease of obtaining high molecular weight DNA from other organisms, entomophthoralean fungi are recalcitrant to standard DNA extraction procedures. The reasons for this are not yet clear, but, along with the relatively large sizes of entomophthoralean genomes, they have stymied sequencing and assembly efforts within this fungal order. Our success came from having on-demand access to an *E. muscae* culture in the lab and to using a modified extraction protocol (see Materials and methods), which could potentially be applied to other entomophthoralean fungi for future genomic work.

While our analysis suggests that this new genome is fairly complete (81.3% complete BUSCOs), we observed a high level of duplicate BUSCOs. Our assembly is decidedly not fully haploid (Fig. S1A,B). Instead, our data are consistent with duplicate BUSCOs arising from heterozygosity in coding regions. If this is the case, we have overestimated the size of the genome by about 20 Mb (2%). The prevailing thinking in the field is that entomophthoralean fungi are haploid (*Humber, 2016*), but the ploidy of *E. muscae* is still not fully resolved. Our observation is reminiscent of the functional diploidy previously reported for another *E. muscae* isolate (*De Fine Licht et al., 2017*).

### A proliferation of transposable elements within Entomophthorales

Given how little is known about the biology of Entomophthorales, we adopted a comparative approach for our initial analysis of the *E. muscae* genome, sourcing available transcriptomic and genomic data from species within this fungal order. The inferred phylogenetic relationship between these species, based on conserved protein sequences (*Figure 1C*), implied that *Conidiobolus* is paraphyletic. This is consistent with recent taxonomic revisions of that genus (*Gryganskyi et al., 2022*; *Nie et al., 2020*).

The second smallest genome (after *Conidiobolus*) in our comparative data set, was *Z. radicans*, which, at 655 Mb, is still >15-fold higher than the mean fungal genome size of all 6144 genomes considered in *Figure 2A* (42.7 Mb). The size of these genomes is primarily driven by a prolific expansion of repeat elements, the most numerous of which are Ty3 retrotransposons (*Figure 2C*). Fungal Ty3 elements are common, and appear to have undergone independent expansions across many fungal lineages (*Muszewska et al., 2011a*). From the consistent patterns of Ty3 sequence abundance and the estimated divergence time of these sequences (*Figure 2D*), we speculate that Ty3 expansion was a feature of the last common ancestor of non-*Conidiobolus* Entomophthorales: this proliferation was not recent.

One possible avenue leading to such a proliferation could be loss of defenses against repeat accumulation. Fungi use repeat induced polymorphism (RIP) and RNA interference (RNAi) as their two main strategies to combat the accumulation of repeated elements. We detected several homologues of the core RNAi pathway components in each of our fungal species: Dicer, Argonaute, and RNA-dependent polymerase (*Figure 2G*). This analysis suggests that RNAi pathways are intact and potentially function in a similar manner to other fungi within Entomophthorales, and may even indicate an expansion of this defense mechanism to counteract TE proliferation.

While RNAi occurs across Fungi, RIP has only been observed in Dikarya. Our analyses suggest that RIP may be active in fungi outside of Dikarya (*Figure 2E and F*), but there are major caveats to this finding. We detected DNA methyltransferases with shared domain architecture as RID, a key enzyme for RIP function, but this should be validated with deeper analyses and experimental characterization of genes in these orthogroups. Host population dynamics limit opportunities for pathogen sexual recombination temporally and spatially, and entomophthoralean fungi are hypothesized to reproduce almost entirely asexually (*Humber, 2016*). If this is the case, these cells would not undergo meiosis, which is the cell-cycle stage where RIP occurs in fungi in the subkingdom Dikarya. Parasexuality (i.e. heterokaryons formed via arbitrary fusion of genetically distinct cells) may represent another opportunity for RIP. If parasexuality occurs in entomophthoralean fungi, a suitable moment might be in the protoplast stage, when cells lack cell walls and could meet *in insecta* after successfully infecting the host. Protoplast production is a necessary step for parasexual fusing in the laboratory setting. However, the existence of parasexuality has been questioned in this group of fungi, as cell-to-cell fusion has only been observed to occur between genetically identical hyphal bodies prior to resting spore formation (*Humber, 2016*). Such fusion may still provide an opportunity to silence TEs that accumulate during

primarily asexual development. However, parasexuality may also provide an opportunity for TEs to spread among heterokaryotic nuclei, as has been suggested in arbuscular mycorrhizal fungi (*Yildirir et al., 2020*) and may be particularly amenable for transposons with RNA intermediates (such as Ty3). In support of the presence of RIP in entomophthoralean fungi, all fungi in our set possess genes encoding DNA methylase domains (DNA_methylase: PF00145). If these genes are indeed found to mediate RIP, it would position *E. muscae* as a model for investigating meiosis-independent RIP.

Rust fungi parasitize plants (e.g. *Phakospora pachyrhizi*, *Astropuccinia psidii*; *Figure 2A and B*) and also have very large genomes as a result of rampant TE proliferation. Recent analysis of three isolates of *Phakospora pachyrhizi* found that, similar to Entomophthorales, Ty3 elements comprise 43% of the genome (*Gupta et al., 2023*) and the expansion of these elements was predicted to coincide with radiation of host species. We do not yet know if TEs in *P. pachyrhizi* and the family Entomophthoraceae expanded via similar mechanisms. The large genomes in these clades are puzzling given that evolution tends to reduce rather than expand parasitic genomes (*Wolf and Koonin, 2013*). TE expansions may have been triggered by a variety of biotic and abiotic factors and underlie evolutionary choice points (*Belyayev, 2014*). But what factors would allow an enlarged genome to persist over evolutionary time? We hypothesize that an enlargement of the non-coding genome may set the stage for evolution of novel functionalities, and, in turn, specialization for different hosts and speciation. Rust and entomophthoralean pathogens, which are both marked by Ty3 element proliferation, are known for their extreme degree of host specialization (*Sacco and Hajek, 2023*). In the absence of frequent sexual recombination, the flexibility permitted by an outsized 'genomic canvas' may have been favored by selection and maintained to the present day. Work addressing the basis of TE proliferation across divergent lineages is needed to better understand the evolutionary pressures driving genomic enlargement and maintenance of giant genomes. Considering the observed Ty3 proliferation in Entomophthorales is not recent (*Figure 2D*), we can theorize a trajectory to the observed TE-inflation and high degree of host specialization in entomophthoralean fungi: first, a transition away from frequent sexuality to infrequent parasexuality/sexuality (lowering TE defenses), then a proliferation of TEs (generating raw materials for adaptation) and finally a slow, parasexual cessation of TE bursts with selection on genes involved in host-specific infection (strengthening the degree of host specialization).

## Entomophthorales share the goal of converting insect tissue to energy; precise approaches vary

Analyses of protein domains (Pfam, CAZy, and MEROPS) on our core set of entomophthoralean genomes (plus two available transcriptomes: *S. castrans* and *P. formicae*) revealed several functions enriched in the predicted *E. muscae* proteome (*Figure 3A*, *Figure 3—figure supplement 2*). Many of the associated proteins are predicted to be secreted and likely metabolize host tissues (lipases, chitinases, insulinases, etc). In addition, trehalase enzymes which act upon the primary sugar of hemolymph (*Thompson, 2003*), were found to be expanded in *E. muscae*, providing further evidence that this fly pathogen has evolved to efficiently process host tissue. As more entomophthoralean fungi are sequenced, the enzymatic repertoire of these highly specialized pathogens will provide insights into the nutritional needs of these fungi, which may vary with differences in tissue composition of their hosts.

Beyond these protein domain and enzyme analyses, we were very interested in comparing potential circadian machinery among entomophthoralean fungi as, for many of these pathogens, infection and behavioral manipulation of the host follows a strict daily timing (*Dustan, 1924*; *Elya et al., 2018*; *Hajek and Harris, 2023*; *Krasnoff et al., 1995*; *Milner et al., 1984*; *Nielsen and Hajek, 2006*; *Pickford and Riegert, 1964*). How this timing is achieved is not understood. One study suggests that timing of death of house flies killed by *E. muscae* is not dictated by host circadian machinery, instead suggesting that a fungal clock drives the timing (*Krasnoff et al., 1995*). In *N. crassa*, the primary fungal model system for circadian biology, two key genes in maintaining circadian rhythms are *frequency* (*frq*) and the blue-light sensitive *white-collar 1* (*wc-1*). While we found many genes encoding light-sensitive protein domains, including several *wc-1* homologs, we did not find homologs of *frq* within our entomophthoralean fungi. However, absence of *frq* does not preclude a functional clock in these fungi; many fungi that lack a *frq* homolog have clear circadian phenotypes under free-running conditions (e.g. Mucoromycotina fungus *Pilobolus sphaerosporus* and ascomycetes *Aspergillus flavus*,

*Saccharomyces cerevisiae* and *Cercospora kikuchii*), indicating that there are *frq*-independent mechanisms for keeping time within Fungi that have yet to be discovered (***Montenegro-Montero et al., 2015***; ***Salichos and Rokas, 2010***). In addition, the ascomycete plant pathogen *Verticillium dahliae* has a homolog of *frq* that does not cycle over circadian time (***Cascant-Lopez et al., 2020***). Notably, *frq* genes have only been observed in Ascomycota (except for the subphylum Saccharomycotina), so it is unsurprising that entomophthoralean fungi would lack homologs of this gene (***Montenegro-Montero et al., 2015***; ***Salichos and Rokas, 2010***).

As a complementary approach to our domain-based analysis, we also categorized genes from our core genomic set (*E. muscae*, *Entomophaga maimaiga*, *Z. radicans,* and *N. thromboides*) into putative orthologous groups (OGs; ***Figure 2E–G***). A plurality of OGs contained genes from all four species (40.2%). The next most common species composition of OGs was *E. muscae* and *E. maimaiga* (10.8%), followed by *E. muscae*, *E. maimaiga,* and *Z. radicans* (7.5%). Each species also possessed species-specific OGs, with *E. muscae* possessing the most. An enrichment analysis of Pfam annotations for *E. muscae* genes that were assigned to core OGs revealed that many processes that may be host-specific (e.g. chitin recognition, fatty acid, protein and sugar utilization) are underrepresented within the core set (***Figure 3—figure supplement 3***). Conversely, genes predicted to serve in signaling pathways (e.g. kinases, pleckstrin homology domains) and basic cellular metabolism (e.g. transcription, translation, and respiration) are overrepresented among *E. muscae* genes assigned to core OGs (***Figure 3—figure supplement 3***). All told, these results are consistent with specialization in utilizing host tissues as a key driver of species-level differences in these fungi.

We also searched for gene-family-level expansions not just within Entomophthorales (***Figure 4A***), but across diverse fungal lineages that include other insect pathogenic species (***Figure 4B***). We observed gene family expansions related to host tissue utilization, some unique to Entomophthorales (e.g. lipases, acid trehalases) and others shared across entomopathogens of different phyla (e.g. peptidases). This partial overlap indicates that there are common themes among insect fungal entomopathogens in conquering their hosts as well as divergent strategies within distinct fungal lineages.

## Entomophthoralean fungi likely produce secondary metabolites

Our analysis of predicted secondary metabolites within insect fungal entomopathogens suggests that Entomophthorales may produce non-ribosomal peptides, siderophores and terpenes. A recent report demonstrating the presence of several terpenoid compounds in *E. muscae*-killed house flies (***Naundrup et al., 2022***) suggests that this prediction likely missed important classes of metabolites. Current models predict that the quantity and diversity of secondary metabolites is lower in entomophthoraleans than for ascomycete entomopathogens (***Figure 4C***), but this conclusion has several caveats. At a first glance, our predictions seem consistent with Entomophthorales following a primary growth, rather than a primary toxin-producing, infection strategy (as proposed for certain generalist *Metarhizium* species ***Kershaw et al., 1999***), and with the long-held dogma that zygomycete fungi produce few secondary metabolites (***Bushley and Turgeon, 2010***; ***Voigt et al., 2016***).

However, secondary metabolite cluster analysis methods have been developed almost exclusively using data from Ascomycota and Basidiomycota. Thus, the lack of predicted secondary metabolites in Entomophthorales could be due to prediction methods not well suited to function outside of Dikarya. Recent evidence for secondary metabolite gene clusters in Mucoromycota opposes the long-held view that zygomycete fungi do not produce these compounds (***Koczyk et al., 2021***; ***Voigt et al., 2016***) and suggests that methods to detect metabolites and the enzymes that synthesize them need to be adjusted for non-Dikarya fungi. Indeed, a recent study in *Massospora* experimentally detected several secondary metabolites that, based on current bioinformatic models of biosynthesis, should not exist in this species (***Boyce et al., 2019***). In addition, because Zoopagomycota are so uncharacterized with respect to secondary metabolites, it is possible that the apparent lack of standard metabolite classes is accurate, but we failed to identify metabolite clusters that are unique to this phylum and not yet described. We have much to learn about secondary metabolism beyond Dikarya and future efforts should prioritize exploring metabolism of these taxa for the potential discovery of novel metabolic gene cluster architecture or novel metabolite synthesis pathways.

## Potentially unique features of *E. muscae* biology

From analysis of Pfam, MEROPS and CAZy domains, we found that *E. muscae* has a handful of protein domains unique within the Entomophthorales species we analyzed. The most numerous of these was Peptidase A2B, which is a protease family involved in processing retrotransposon Ty3 polyprotein into its component parts (*Kirchner and Sandmeyer, 1993*). Perhaps this reflects a species-specific transposon for *E. muscae*. However, given the proliferation of Ty3 elements among family Entomophthoraceae fungi in our dataset, we expect that the functional activity of this domain is not restricted to Peptidase A2B. Importantly, we did not find any domains that were missing from *E. muscae* but enriched in other fungi in our core set (*Figure 5B*, *Figure 5—figure supplement 1C*). Together, these observations are consistent with *E. muscae* having diverged from other entomophthoralean fungi by expansion of existing domain families rather than loss of particular domains.

As an additional approach to identifying unique gene functions in *E. muscae*, we also looked at orthogroups unique to this species, from either the total proteome or the secreted proteome (*Figure 5C and D*). M16 peptidases (domains PF00675 and PF05193) were found to be enriched across all proteins as well as within proteins predicted to be secreted. M16 peptidases are metalloendopeptidases with family members that cleave N-terminal targeting peptides that direct proteins to their destination subcellular compartments (*Rawlings et al., 2018*). Recent work in the the protozoan *Cryptosporidium parvum* implicated insulinase-like proteases (which are also members of the M16 family) in early infection (*Zhang et al., 2019*), perhaps suggesting a role for these proteins in early development of *E. muscae*.

Based on our domain enrichment analysis, we expect that many of the unique secreted proteins are involved in the metabolism of fly host tissues. Depending on the host range specificity of this study's *E. muscae* isolate, these genes could encode proteins for metabolizing or recognizing *Drosophila* macromolecules specifically. Looking at just secreted proteins, we saw enrichment for a handful of catabolic domains including peptidase inhibitor I9 (PF05922), a constituent of subtilisins (*Arnesen et al., 2018*; *Muszewska et al., 2011b*). This observation is consistent with work finding unique subtilisin-like serine proteases in the entomopthoralean fungi *E. muscae*, *C. incongruus* and *P. formicae* and suggests further subspecialization of these enzymes at the species level (*Arnesen et al., 2018*). We also observed an enrichment for chitin recognition protein (PF00187) in the secreted proteome. Both the fly host and *E. muscae* produce chitin, so these proteins could function in either host recognition or in masking *E. muscae* from being recognized by the host immune system upon entry (*Cen et al., 2017*).

An enrichment for the collagen triple helix repeat domain (PF01391) caught our attention. In the generalist fungal pathogen *Metarhizium anisopliae*, a collagen-like protein (Mcl1) that is expressed immediately after invasion of the host hemolymph helps fungal cells avoid detection by the host immune system (*Wang and St Leger, 2006*). In a similar vein, we wonder if *E. muscae*'s unique collagen triple helix domain may play a role in evading the host immune system.

We also were surprised to observe an enrichment in berberine-like proteins (PF08031) among secreted *E. muscae*-specific OGs. Berberine-like proteins are involved in the synthesis of isoquinoline alkaloids, a class of compounds that includes the analgesics morphine and codeine, and usually require FAD as a cofactor (*Kutchan and Dittrich, 1995*). Intriguingly, we also observed an enrichment of FAD binding domains (PF01565). Isoquinolines have been most frequently discovered in plants, but recent work in *Aspergillus fumigatus* discovered that an orphan metabolic gene cluster containing small NRPS-like genes encodes enzymes capable of producing a novel isoquinoline (*Baccile et al., 2016*). Isoquinolines have wide-ranging effects (*Khan and Suresh Kumar, 2015*), and some isoquinolines have been shown to inhibit the innate immune response. Future work leveraging high-sensitivity metabolomics methods will be needed to determine if isoquinolines are produced by *E. muscae* and, if so, what role they may play in infection and in altering host behavior.

## Recommendations for Entomophthora species identification

Historically, *Entomophthora* spp. were identified using a combination of morphology, host species, and/or location (*MacLeod et al., 1976*; *MacLeod and Müller-Kögler, 1973*). However, in recent years it has come to light that these features may be unreliable. For example, conidial size of members of the EMSC changes when infecting host flies different from the original source host (*Jensen et al., 2006*), meaning that this feature is not, on its own, reliable for identifying the fungal species. Such

differences in conidial size across insect hosts have also been observed for another closely allied member of the Entomophthoraceae, *Massospora levispora* (*Macias et al., 2020*). Another study examining haplotype diversity in *E. muscae* strains from a single epizootic infecting two fly species in NC, USA uncovered two genetically distinct subpopulations, both with nearly identical primary conidial sizes and numbers of nuclei per conidium (*Gryganskyi et al., 2013*).

To explore the utility of DNA sequence data (ITS +28 S) for identification purposes in the EMSC, we assessed the congruence of available sequence and morphology data for EMSC isolates. We recovered two well-supported clades (both in the single gene trees and the concatenated tree) that contain strains previously identified as *E. muscae, E. ferdinandii*, and *E. scatophagae* (*Figure 6A–C*). Using morphological characters, we found that the number of nuclei in the primary conidia contained almost no phylogenetic signal among members of the EMSC based on the strains we assessed (*Figure 6B*). Furthermore, the dimensions of the primary conidia were largely overlapping among nearly all *Entomophthora* species examined, with the exception of *E. ferdinandii* and *E. schizophorae* (*Figure 6C*). Interestingly, dimensions of primary conidia for our *E. muscae* strain (ARSEF 13514) overlapped with measurements for *E. ferdinandii* and outside the reported measurements for *E. muscae* despite occupying the same well-supported clade as strains whose spore measurements agreed with *E. muscae*.

Taken together, these results indicate that the morphological characters examined here, which are traditionally used to identify Entomophthorales, are incongruent with molecular data especially within the EMSC. However, this disagreement between morphology and sequence data could reflect (1) the paucity of character data, (2) insufficient taxon sampling that failed to capture the full range of morphological and genetic variation, and/or (3) the presence of misidentified specimens in collections and studies that are potentially masking the presence of morphologically cryptic species (i.e. if researchers relied on nuclei number and conidia measurements to identify their specimens in the absence of DNA sequence data, then any morphology data they report for the species could include data from multiple species). While ITS and 28 S loci may have insufficient phylogenetic signal to fully resolve these species boundaries especially within the EMSC, resolution may be achievable with the addition of other loci. In addition, the evolutionary history of this group may include a combination of introgression and rare sexual recombination events that have softened species boundaries within the EMSC, though data supporting this are currently limited (*Gryganskyi et al., 2013*). Overall, the EMSC remains monophyletic, but a comprehensive taxonomic revision is needed to define species within the clade.

Many strains of EMSC exist in fungal collections (ARSEF in particular) that have no associated sequence or morphology data. We propose that for each individual specimen, whether novel or from a collection, researchers minimally generate ITS and 28 S sequence data and collect morphological data (specifically the length, width, and number of nuclei within primary conidia), so that future research can assess these features in combination to define the species boundaries of this group. We encourage researchers to further deposit voucher specimens, live cultures and/or DNA with fungal collections (e.g. ARSEF) so that future researchers can return to the original sample to examine traits or generate more sequence data if needed. Likewise, historic studies based solely on morphology from herbarium specimens may need to be resampled for both DNA sequencing and morphology.

## What's next?

The work presented here is one of the first steps in exploring Entomophthoralean genomes. We hope that the data and questions posed herein can serve as a springboard for generating and testing new hypotheses about these organisms. *Entomophthora muscae* has been under scientific scrutiny since its description in 1855, yet, even under the new light of our genomic investigation, this fungus remains 'one of the strangest and most interesting apparitions' (*Elya and De Fine Licht, 2021*) with clear avenues to enrich our understanding of myriad biological phenomena.

## Materials and methods

### E. muscae culture and DNA extraction

*E. muscae* (ARSEF 13514) was isolated for in vitro growth from a single sporulating cadaver in 60 mm sterile petri dishes using the ascending sporulation method (*Hajek et al., 2012*). All cultures were grown in supplemented Grace's Insect Media (ThermoFisher Scientific #11605094) with 5% fetal

bovine serum added (Thermo Fisher Scientific #10437010) in volumes of 20 mL in 25 cm² vented tissue culture flasks (Corning #353014). Cultures were grown without shaking, with flasks laid on the long edge to maximize surface oxygen exchange, in the dark at room temperature (19–21 C). DNA was isolated from a log-phase culture via methods of *Elya and Lee, 2022*. DNA quantity, quality, and size distribution were assessed by Qubit dsDNA HS Assay kit (Thermo Fisher Scientific #Q32851), Nano-drop spectrophotometric analysis (Thermo Fisher Scientific #ND2000) and DNA Genomic ScreenTape analysis (Agilent TapeStation #5067–5365 and #5067–5366) following manufacturer's protocols.

## Genomic sequencing and assembly

Three µg of genomic DNA were used to prepare two Oxford Nanopore DNA libraries, one sheared to 20 kb and one unsheared (45–50 kb) and sequenced sequentially on the Oxford Nanopore PromethION platform by the The Bauer Core Facility at Harvard University (Cambridge, MA) to 82 x coverage. Reads are accessible through the NCBI Sequence Read Archive (SRA) via accession number SRR18312934. Nanopore reads that passed filtering were assembled using Flye v2.8.3 (*Kolmogorov, 2021*; *Kolmogorov et al., 2019*). The resultant assembly was polished with the same Nanopore reads used for assembly using Medaka v1.2.6 (*Oxford Nanopore Technologies, 2021*).

The Flye +Medaka assembly was further scaffolded with 10 x genomic sequence library reads from the UCB isolate of *E. muscae* (*Elya et al., 2018*) deposited in NCBI SRA via accession number SRR18312935. This was accomplished by demultiplexing with longranger v2.2.2 (10x genomics), reads mapped with bwa-mem2 (*Vasimuddin et al., 2019*), followed by assembly scaffolding with Tigmint v1.2.2 (*Jackman et al., 2018*) and ARCS 1.2.1 (*Yeo et al., 2018*).

## Genome annotation

This updated *E. muscae* UCB genome was annotated with Funannotate v1.8.9 (*Palmer and Stajich, 2020*) employing the default parameters, which automates processing of RNAseq to produce transcripts, gene prediction with evidence from the transcripts and alignments of proteins, and construct consensus gene models from multiple gene prediction evidence, refine gene models for alternative splicing prediction based on the RNA-seq, and produce product function annotation based on protein sequence homology to databases of domains and annotated proteins. To annotate the genome, first repetitive sequences were masked by building a species-specific repeat library with RepeatModeler v2.0.1 (*Flynn et al., 2020*). These putative repeat families were further screened manually to remove likely protein-coding genes based on Diamond BLASTX v2.0.8 (*Buchfink et al., 2021*) searched against SwissprotDB v2021_04 (*UniProt Consortium, 2021*) and removing sequences that matched clear non-repetitive but multicopy gene families. The curated repeat library was combined with the RepBase (*Bao et al., 2015*) library of fungi repeats with RepeatMasker v4.11 (*Smit et al., 2013*) to softmask the genome before gene prediction to avoid over predicting transposons as host genes.

To train gene predictors and support gene models, RNA-seq data from previously deposited sequence data in NCBI SRA accession ERR1022665 were used as informant data in the annotation process. Briefly, the RNA-seq processing used Trinity v2.11.0 (*Haas et al., 2013*) in Genome Guided mode which aligned reads to the genome with Hisat2 v2.2.1 *Kim et al., 2019* followed by targeted transcript assembly with the Trinity pipeline on reads clustered to distinct genomic locations. The constructed assemblies total 63,231 which reflect partial transcripts and potential alternative splicing. These sequence assemblies were aligned to the genome with PASA (*Haas et al., 2008*) which uses the splice-aware aligner GMAP (*Wu et al., 2016*) and additional software to produce gene models from transcript data. This produced 67,902 gene models. The transcripts with full-length Open Reading Frames as scored by Transdecoder v5.5.0 (*Haas et al., 2013*) were kept as high quality models for gene prediction training. A total of 2653 full-length PASA-derived gene models were used as a training set to *ab initio* predictors SNAP v2013_11_29 (*Korf, 2004*) and AUGUSTUS v3.3.3 (*Stanke et al., 2008*). In addition, the tools GeneMark-ES v4.62 (*Ter-Hovhannisyan et al., 2008*) and GlimmerHMM v3.0.4 (*Majoros et al., 2004*) were run after these tools performed self-training. Gene models were also predicted by CodingQuarry (*Testa et al., 2015*) using the transcript alignments as exon hints. Exon evidence was also generated by DIAMOND alignment of SwissprotDB proteins and polished by Exonerate v2.4.0 (*Slater and Birney, 2005*). The transcript and protein-based hints were provided to GeneMark, SNAP, and AUGUSTUS for evidence-guided gene prediction. EVidenceModeler v1.1.1 (*Haas et al., 2008*) generated consensus gene models in Funannotate using its

default evidence weights. tRNA genes were predicted by tRNAscan-SE v.1.3.1 (*Lowe and Eddy, 1997*). Putative protein functions were assigned to genes based on sequence similarity to the Inter-ProScan v5.51–85.0 (*Blum et al., 2021*; *Jones et al., 2014*), Pfam v35.0 (*Mistry et al., 2021*), Eggnog v2.1.6-d35afda (*Cantalapiedra et al., 2021*), dbCAN2 v9.0 (*Huang et al., 2018*), and MEROPS v12.0 (*Rawlings et al., 2018*) databases relying on NCBI BLAST v2.9.0+ (*Camacho et al., 2009*) and HMMer v3.3.2 (*Eddy, 2011*). Predicted secreted genes and transmembrane domains were annotated with Phobius (*Käll et al., 2004*) and SignalP v5.0b (*Almagro Armenteros et al., 2019*). A total of 39,711 gene models comprising 42,665 predicted proteins with alternative splicing isoforms, and 793 tRNAs were predicted. Pipeline for annotation is archived in the github repository https://github.com/zygo-life/Entomophthora_muscae_UCB (copy archived at *zygolife, 2023*) and Zenodo archive (https://doi.org/10.5281/zenodo.8339801).

## Fungal datasets

A comprehensive overview of all of the fungal datasets used in this paper and their associated figures is provided in *Supplementary file 1b*.

Transcriptomic data for *Strongwellsea castrans* sensu *lato* were generated from a cabbage fly (*Delia radicum*) caught in a cabbage field (farm name: Sørisgård, Latitude: 55.823706, Longitude 12.171149, date: 04/09/2013). The fly contained a clearly visible large hole on the side of the abdomen characteristic of *S. castrans* infection and was kept alive for a few hours until being snap-frozen in liquid nitrogen. The infected fly was ground to a fine powder in liquid nitrogen before extracting total RNA using a Plant RNEasy Kit (QIAGEN #74904) following manufacturer's instructions. Total RNA was sequenced using Illumina HiSeq 2000 technology and TRUseq library building by Beijing Genomics Institute (BGI-Europe, Copenhagen, Denmark), and transcriptome sequences were assembled in de novo mode using Trinity v2.11.0. Raw reads are deposited at the European Nucleotide Archive (ENA) with accession number: ERR12023556. Assembled transcripts are available in Zenodo archive under Supporting_data folder (https://doi.org/10.5281/zenodo.8339798).

## Phylogeny of species

Phylogenetic relationships of species were determined by identifying conserved proteins from the protein translation of predicted genes or transdecoder ORFs from the transcriptome-only sampled species (*P. formicae* and *S. castrans*) using the PHYling v1.1 (*Stajich and Tsai, 2023*) BUSCO / OrthoDB fungi_odb10 marker set. Briefly, the pipeline searches for conserved, generally single copy proteins via HMMer (*Eddy, 2011*) searches and builds individual and concatenated protein alignments. The phylogenetic tree was constructed from the concatenated alignments with FastTree v2.1.11 (*Price et al., 2009*) using the '-lg -gamma' parameters. Comparison of this topology matched previous published relationships of these lineages (eg *Boyce et al., 2019*; *Wang et al., 2023*) which did not justify further exploration with additional likelihood or Bayesian methods.

## BUSCO analysis

Genome completeness was computed with BUSCO v5.2.2 (*Manni et al., 2021*) using the -m genome mode on the masked assembly employing either fungi_odb10 or eukaryota_odb10 marker sets. The analysis was also performed on the predicted proteins using the -m protein mode.

## Genome size and gene counts

Data used to generate *Figure 2A and B* were compiled from NCBI, MycoCosm, the Fungal Genome Size Database (zbi.ee/fungal-genomesize/) and (*Mohanta and Bae, 2015*) and are available as *Supplementary file 2*. In some cases, multiple strains of the same species have been sequenced.

## Repeat analysis

To identify repetitive sequences in the entomophthoralean genomes we used RepeatModeler v2.0.1 (*Flynn et al., 2020*) to develop a de novo library of elements and classify their likely lineage using default parameters and LTR finding with LTRStruct option. The elements were classified by similarity to RepBase v20170127 (*Bao et al., 2015*) using the RepeatClassifier component of RepeatModeler. The de novo repeat library was combined with all identified fungi repeats in RepBase to produce a composite library to identify repetitive regions in the genomes with RepeatMasker v4.1.1 (*Smit et al.,*

*2013*) genome masking and transposon exploration. The repeat landscape plots were generated with the RepeatMasker script createRepeatLandscape.pl using default options.

To test for the presence of Repeat Induced Point Mutation patterns in the genome, a RIP index was calculated for 1 kb windows with 500 bp offsets using the composite index (*Lewis et al., 2009*). Summary statistics were calculated to score the number of windows where the composite RIP index was greater than 0 to summarize the % of the genome RIPped. Scripts and summarized values from each genome are part of the https://github.com/zygolife/Emuscae_Comparative repository (copy archived at *Stajich and Lovett, 2023*) and archived in Zenodo (https://doi.org/10.5281/zenodo.8339798) in the 'comparative/RIP' folder.

## Domain and comparative analysis

Domain analysis for seven entomophthoralean proteomes (*E. muscae*, *Entomophaga maimaiga*, *Z. radicans*, *P. formicae*, *S. castrans*, *N. thromboides*, and *C. coronatus*) was limited to MEROPS, CAZy and Pfam annotations. These annotations were completed according to a pipeline identifying domains by sequence similarity and motif searches https://github.com/stajichlab/Comparative_pipeline (*stajichlab, 2018*). The identified domain counts per species aggregated as total counts or total number of unique genes with a domain were compiled into a single table for comparison between species.

In RStudio v1.4.1717 (*RStudio Team, 2022*), a custom R v4.1.0 (*R Development Core Team, 2021*) script for enrichment analysis is available in https://github.com/zygolife/Emuscae_Comparative repository and archived (https://doi.org/10.5281/zenodo.8339798), which relies on a number of R packages: tidyverse v1.3.1 (*Wickham et al., 2019*), fmsb v0.7.2 (*Nakazawa, 2022*), grid (an R base package), gridExtra v2.3 (*Auguie, 2017*), ComplexUpset v1.3.3 (*Krassowski et al., 2022*), UpSetR v1.4.0, ggforestplot v0.1.0 (*Scheinin et al., 2020*), broom v0.7.12 (*Robinson et al., 2023*) and viridis v0.6.2 (*Garnier et al., 2021*). For each domain type, the number of accessions containing each domain was counted. Domains that were present in at least two genomes were used for pairwise enrichment analysis among genomes. For each domain type, enrichment of counts was calculated using fmsb:: pairwise.fisher.enrichment with Bonferroni correction and a p-value threshold of 0.01 for significance. For each genome, the number of genes found to be significantly different were counted, and the median count for each domain was used to estimate the fold compared to median and a direction (i.e. up or down) compared to median.

Count tables were used to determine lists of domains present within each genome for set analysis including UpSet plots (ComplexUpset and UpSetR) and analysis of unique/missing genes. A second set analysis was conducted with Pfam domains including an additional *E. muscae* proteome from RNAseq data (EMU-T) generated from the same strain (NCBI GEO GSE111046).

Putative circadian genes were surveyed using Pfam domain annotations. The following domains were used for this survey: 7tm_1 (PF00001), FRQ (PF09421), GATA (PF00320), PAS_3 (PF08447), PAS_9 (PF13426), PAS (PF00989), Bac_rhodopsin (PF01036), GpcrRhopsn4 (PF10192), DNA_photolyase (PF00875), FAD_binding_7 (PF03441), PHY (PF00360), and GAF (PF01590). These were used to filter single-domain circadian candidates from each genome for our domain analysis. Additionally, we considered the expected domain pattern of known circadian genes to identify curated candidates for FRQ (one FRQ domain), WC1 (one GATA and one PAS_3 and one PAS_9), WC2 (one GATA and one PAS_3 and no PAS_9) and 7tm_1 (rhodopsin; one 7tm_1 domain).

RNAi pathway candidates were similarly surveyed using Pfam domain annotations. Domains included in this survey were: RdRP (PF05183), Dicer_dimer (PF03368), PAZ (PF02170), Piwi (PF02171), Ribonuclease_3 (PF00636), and DEAD (PF00270). Identified candidates containing surveyed domains, but we further employed an expected domain pattern to curate candidates for RNAi pathway proteins: RDRP (containing RdRP domain), Dicer (containing Dicer_dimer), Ago (containing both PAZ and Piwi), Dicer_Alt (containing Ribonuclease_3 and either DEAD or PAZ).

## Orthogroup analysis

Orthologous genes were identified between the compared entomophthoralean species by first taking the longest peptide for each gene to avoid including alternative spliced isoforms in the analyses. Proteins were clustered with OrthoFinder v2.5.2 (*Emms and Kelly, 2015*) using Diamond v2.0.6.144 (*Buchfink et al., 2021*) with the ultra-sensitive parameter.

Gene expression was estimated by aligning a pooled set of 27 in vivo samples whole female fruit flies that had been exposed to *E. muscae* 24–120 hours previous from NCBI #GSE111046 *Supplementary file 1d* to the annotated assembly using Kallisto (v.0.46.1). Genes with fewer than five estimated counts across this pooled set were filtered out of the potentially species-specific gene set (a collection of genes that either failed to cluster in any orthogroup across *E. muscae*, *E. maimaiga*, *Z. radicans*, and *N. thromboides* or appeared in an orthogroup that was exclusively populated by *E. muscae* genes). Pfam enrichment of orthogroups was assessed via Fisher's exact test (Matlab function *fishertest*).

Orthogroups were used to identify methyltransferases that share RID domain architecture. RID is annotated with multiple Pfam DNA_methylase (PF00145) domains. RID candidate proteins, containing more than one DNA_methylase, were identified with the same approach we used to identify circadian protein candidates. Where necessary, homologs that corresponded with genomes used in orthogroup analysis were identified via BLAST. All proteins within orthogroups that contained RID candidates (OG0001715 and OG0003300) were grouped in Geneious Prime (2021.2.2) and aligned using MAFFT (7.450; BLOSUM62) with the sequence of RID (AAM27408.1) from *N. crassa*. This alignment was used to produce a protein tree using RAxML (8.2.11; GAMMA BLOSUM62 with 1000 bootstraps). This tree was manually rooted to separate orthogroups, maintaining the clade containing RID and OG0001715. The basal *E. muscae* protein DSO57_1016266-T1 is apparently partial (140 aa) compared to other *E. muscae* proteins in OG0001715 (both are >1200 aa).

## Gene family expansion analysis

Protein sequences containing extracellular trehalase enzymes (PF01204) were retrieved from the data and aligned using MAFFT, which was used to calculate a maximum likelihood phylogenetic tree with RAxML using default parameters. In addition, total number of genes and number of genes encoding the PFAM domains Lipases (Lipase_3), Subtilisin-like serine peptidases (Peptidase_S8), Trehalases (Trehalase), Trypsins (Trypsin), and Chitinases (Glycohydro_18) in representative fungal species of Zoopagomycota and Ascomycota were retrieved from JGI Mycocosm and compared to our data of *E. maimaiga* (EMA), *Z. radicans* (ZRA), *N. thromboides* (CTH), *C. coronatus* (Conco1), and *Basidiobolus meristosporus* (Basme2finSC).

## Antismash secondary metabolite prediction

To examine secondary metabolism potential for the *E. muscae* genome and its relatives, fungiSMASH v6.0.0 of antiSMASH (*Blin et al., 2021*) was run as part of the funannotate genome annotation steps with the options '--taxon fungi --genefinding-tool none --fullhmmer --clusterhmmer --cb-general --cassis --asf --cb-subclusters --cb-knownclusters'. The predicted clusters were incorporated into the genome annotation submitted to GenBank.

## Mining and analysis of ribosomal RNA sequences and morphological data for Entomophthora muscae species complex isolates

DNA sequence data for members of the *E. muscae* species complex and other *Entomophthora* spp. were mined from NCBI GenBank. Novel Sanger sequence data were also generated in support of this analysis, which has been deposited in NCBI (i.e. ARSEF_13514, ARSEF_6918, SoCal_c1 and LTE_c1). DNA was extracted from individual EMSC-killed cadaver drosophilids collected across California as per *Elya et al., 2018* using a QIAamp DNA Micro Kit (QIAGEN #56304). Extracted DNA was used to template PCR reactions with GoTaq polymerase (Promega #M3001) to amplify ITS (primers emITS-1: TGGTAGAGAATGATGGCTGTTG and emITS-4:GCCTCTATGCCTAATTGCCTTT) and/or LSU regions (primers LR0R-4:GTACCCGCTGAACTTAAGC and LR3-1:GGTCCGTGTTTCAAGAC) (*James et al., 2006*). PCR reactions were enzymatically cleaned with ExoSAP-IT (ThermoFisher #78201.1 .ML) per manufacturer's instructions and submitted to ELIM (Hayward, CA) for Sanger sequencing with both forward and reverse primers. Consensus sequences were assembled in UGENE (Unipro). Sequences were aligned to assess length and quality, and metadata was compiled to determine which strains had available sequence data and morphological data (*Supplementary file 1c*). Morphological data were collected by cross-referencing *Entomophthora* strain IDs with published peer-reviewed literature as well as unpublished/raw data provided by two of this paper's co-authors. Fourteen strains were chosen for inclusion based on the availability of both DNA sequence data and morphological data.

## *E. muscae* phylogenetic tree

ITS and 28 S sequences were aligned separately using MAFFT (*Katoh and Standley, 2013*) on the Guidance2 server (http://guidance.tau.ac.il/; *Landan and Graur, 2007*; *Sela et al., 2015*), and individual residues with Guidance scores <0.5 were masked (1.6% of all residues for ITS, 0.1% for 28 S). Overall Guidance scores for each locus were 0.955 and 0.995 for ITS and 28 S, respectively (alignments with scores approaching 1.0 have high confidence). Nucleotide substitution models were chosen using corrected Akaike information criterion (AICc) scores in Model Test in MEGA X 10.2.6 (*Kumar et al., 2018*; *Stecher et al., 2020*). Alignments of each individual locus, and a concatenated alignment of the two, were used in a maximum likelihood (ML) analysis (RAxML 8.2.12; *Stamatakis, 2014*) and a Bayesian inference (BI) analysis (MrBayes 3.2.5; *Ronquist et al., 2012*), for a total of six analyses. In brief, for ML analyses, an appropriate model was chosen, partitions were applied (for each locus in the concatenated analysis only), 1000 bootstrap replicates were used, and the best-scoring tree was identified and bootstrapped in a single run. For BI analyses, MrBayes was allowed to select a substitution model for each data set, and rates were set based on results from Model Test. One cold chain and three heated chains were used for each of 2 runs, and the first 25% of generations were discarded as burn-in. Each analysis was set for 1 million generations, and no additional generations were needed because the standard deviation of split frequencies fell below 0.01. Finally, run parameters were checked for convergence in Tracer 1.7.1 (*Rambaut et al., 2018*). Trees were viewed and prepared for publication using FigTree 1.4.4 (*Rambaut, 2017*) and Inkscape 0.92.2 (https://www.inkscape.org/). All resulting trees and alignments are available in Zenodo archive (https://doi.org/10.5281/zenodo.8339798).

## Acknowledgements

We are grateful to The Bauer Core Facility at Harvard University for performing library preparation and sequencing of *E. muscae* genomic DNA and for the Bioinformatics team at The FAS Informatics Group at Harvard University for their support in processing and analyzing the resultant data. We also thank the Joint Genome Institute (JGI) for making available the assembly and annotation for the *Entomophaga maimaiga* genome produced under proposal 10.46936/10.25585/60001019 and available at https://mycocosm.jgi.doe.gov/Entmai1. JGI (https://ror.org/04xm1d337) is a Department of Energy User Facility supported by the Office of Science of the U.S. Department of Energy operated under Contract No. DE-AC02-05CH11231. Work to develop the *Entomophaga maimaga* genome was also supported by the Genomics Facility at Cornell University and Dr. J Romero-Severson of Notre Dame University. Finally, kudos to Ryan Bracewell for identifying drosophilids infected with *E. muscae* (SoCal cadaver 1 and LTE cadaver) and providing these specimens for this study.Funding: JES was supported by the National Science Foundation (NSF) (DEB-1441715 & EF-2125066) and the U.S. Department of Agriculture (USDA) (National Institute of Food and Agriculture Hatch projects CA-R-PPA-211-5062-H). Analyses were performed on the UC Riverside High Performance Computing Cluster supported by the NSF (DBI-1429826 & DBI-2215705) and the National Institutes of Health (NIH) (S10-OD016290). AMM is supported by an WVU Outstanding Merit Fellowship. BL was supported by the USDA (USDA-ARS Project 8062-22410-007-000D). BLdB is supported by the Alfred P Sloan Foundation (Research Fellowship), the Esther A and Joseph Klingenstein Fund (Klingenstein-Simons Fellowship Award), the Richard and Susan Smith Family Foundation (Odyssey Award), a Harvard/MIT Basic Neuroscience Grant, the NIH/NINDS (1R01NS121874-01), and the NSF (IOS-1557913). AEJ is supported by the USDA Forest Service (15-CA-11420004-095). HHDFL is supported by a Sapere Aude: DFF-Starting Grant (8049-00086B) from the Independent Research Fund Denmark and a Carlsberg Foundation Young Researcher Fellowship (CF20-0609). CNE is supported by Howard Hughes Medical Institute (Hanna H Gray Postdoctoral Fellowship GT11087).

## Additional information

### Competing interests

Jason E Stajich: Was a paid consultant for Zymergen, Sincarne, and Michroma and is a CIFAR fellow in the program Fungal Kingdom: Threats and Opportunities. The other authors declare that no competing interests exist.

## Funding

| Funder | Grant reference number | Author |
| --- | --- | --- |
| National Science Foundation | DEB-1441715 | Jason E Stajich |
| National Science Foundation | EF-2125066 | Jason E Stajich |
| National Institute of Food and Agriculture | CA-R-PPA-211-5062-H | Jason E Stajich |
| National Science Foundation | DBI-1429826 | Jason E Stajich |
| National Science Foundation | DBI-2215705 | Jason E Stajich |
| National Institutes of Health | S10-OD016290 | Jason E Stajich |
| West Virginia University | Outstanding Merit Fellowship | Angie M Macias |
| United States Department of Agriculture | 8062-22410-007-000D | Brian Lovett |
| Alfred P. Sloan Foundation | Research Fellowship | Benjamin L de Bivort |
| Richard and Susan Smith Family Foundation | Odyssey Award | Benjamin L de Bivort |
| Esther A. and Joseph Klingenstein Fund | Klingenstein-Simons Fellowship Award | Benjamin L de Bivort |
| National Institute of Neurological Disorders and Stroke | 1R01NS121874-01 | Benjamin L de Bivort |
| National Science Foundation | IOS-1557913 | Benjamin L de Bivort |
| USDA Forest Service | 15-CA-11420004-095 | Ann E Hajek |
| Independent Research Fund Denmark | 8049-00086B | Henrik H De Fine Licht |
| Carlsberg Foundation | CF20-0609 | Henrik H De Fine Licht |
| Howard Hughes Medical Institute | GT11087 | Carolyn Elya |

The funders had no role in study design, data collection and interpretation, or the decision to submit the work for publication.

## Author contributions

Jason E Stajich, Conceptualization, Data curation, Formal analysis, Funding acquisition, Investigation, Methodology, Resources, Software, Visualization, Writing – original draft, Writing – review and editing; Brian Lovett, Henrik H De Fine Licht, Data curation, Funding acquisition, Investigation, Methodology, Software, Visualization, Writing – original draft, Writing – review and editing, Conceptualization, Resources; Emily Lee, Formal analysis, Visualization, Writing – review and editing; Angie M Macias, Funding acquisition, Investigation, Methodology, Software, Visualization, Writing – original draft, Writing – review and editing, Conceptualization, Resources; Ann E Hajek, Formal analysis, Software, Resources; Benjamin L de Bivort, Software, Conceptualization, Resources; Matt T Kasson, Data curation, Funding acquisition, Investigation, Methodology, Supervision, Software, Visualization, Writing – original draft, Writing – review and editing, Conceptualization, Resources; Carolyn Elya, Conceptualization, Data curation, Formal analysis, Funding acquisition, Investigation, Methodology,

Project administration, Resources, Software, Supervision, Visualization, Writing – original draft, Writing – review and editing

**Author ORCIDs**
Jason E Stajich ⓘ https://orcid.org/0000-0002-7591-0020
Brian Lovett ⓘ http://orcid.org/0000-0002-5721-7695
Benjamin L de Bivort ⓘ https://orcid.org/0000-0001-6165-7696
Matt T Kasson ⓘ https://orcid.org/0000-0001-5602-7278
Henrik H De Fine Licht ⓘ https://orcid.org/0000-0003-3326-5729
Carolyn Elya ⓘ https://orcid.org/0000-0002-9634-0303

Reviewer #1 (Public Review): https://doi.org/10.7554/eLife.92863.3.sa1
Author response https://doi.org/10.7554/eLife.92863.3.sa2

---

## Additional files

**Supplementary files**
• MDAR checklist

• Supplementary file 1. Summary data for gene counts using different annotation methods, tables of fungal isolates, and RNA-seq datasets used in this study. (a) Variance in predicted gene models using different annotation pipelines does not explain the large gene count predicted in the E. muscae genome. (b) Summary of fungal isolates and data used in this manuscript. (c) Information about strains used in phylogenetic and morphologic studies (related to *Figure 6*). (d) SRA accession numbers of *E. muscae* RNAseq data (NCBI GSE111046) used for pooled expression analysis (related to *Figure 3*).

• Supplementary file 2. Genome sizes and gene counts across Fungi (related to *Figure 2*).

**Data availability**
Genomic sequencing data are available via the NCBI SRA (#SRR18312935). Sequencing reads are available at the NCBI SRA (#SRR18312934). Previously published E. muscae transcriptomic data were accessed from the NCBI SRA (#ERR1022665). Genomic annotation data are deposited at Zenodo (https://zenodo.org/records/8339801). Comparative analyses scripts and data are deposited at Zenodo (https://zenodo.org/records/8339798).

The following datasets were generated:

| Author(s) | Year | Dataset title | Dataset URL | Database and Identifier |
|---|---|---|---|---|
| Stajich JE, Lovett B, Lee E, Macias AM, Hajek AE, de Bivort BL, Kasson MT, Henrik H, de Licht F, Elya C | 2023 | 10X DNA Sequencing of Entomophthora muscae strain UCB | https://ncbi.nlm.nih.gov/sra/SRR18312935 | NCBI Sequence Read Archive, SRR18312935 |
| Stajich JE, Lovett B, Lee E, Macias AM, Hajek AE, de Bivort BL, Kasson MT, de Fine Lich HH, Elya C | 2023 | Genome annotation of Entomophthora muscae strain UCB | https://zenodo.org/records/8339801 | Zenodo, 10.5281/zenodo.8339801 |
| Stajich JE, Lovett B, Lee E, Macias AM, Hajek AE, de Bivort BL, Kasson MT, de Fine Licht HH, Elya C | 2023 | Entomophthora muscae comparative genomics | https://zenodo.org/records/8339798 | Zenodo, 10.5281/zenodo.8339798 |
| Stajich JE, Lovett B, Lee E, Macias AM, Hajek AE, de Bivort BL, Kasson MT, Henrik H, de Licht F, Elya C | 2023 | Nanopore DNA Sequencing of Entomophthora muscae strain UCB | https://ncbi.nlm.nih.gov/sra/SRR18312934 | NCBI Sequence Read Archive, SRR18312934 |

The following previously published datasets were used:

| Author(s) | Year | Dataset title | Dataset URL | Database and Identifier |
|---|---|---|---|---|
| Elya C, Lok TC, Spencer QE, McCausland H, Martinez CC, Eisen M | 2018 | Transcriptomic response of *Drosophila melanogaster* whole bodies or dissected brains to Entomophthora muscae 'Berkeley' | https://www.ncbi.nlm.nih.gov/geo/query/acc.cgi?acc=GSE111046 | NCBI Gene Expression Omnibus, GSE111046 |
| de Fine Licht HH, Jensen AB, Eilenberg J | 2015 | Illumina HiSeq 2000 paired end sequencing | https://ncbi.nlm.nih.gov/sra/?term=ERR1022665 | NCBI Sequence Read Archive, ERR1022665 |

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
