## [Editor Report · eLife assessment]

This **important** study reports on the genome evolution of a poorly studied fungal group. By combining long-read sequencing and different bioinformatic analyses, the authors show that the giant genome of *Entomophthora muscae* expanded due to extensive transposable element activity. The strength of evidence is **compelling** and the authors are to be commended for their multiple comparative analyses of gene content along with transparently written and visualized techniques, data curation, and methods. This paper will be of relevance to fungal biologists as well as to evolutionary biologists interested in the study of genome size dynamics.

---

## [Referee Report · Reviewer #1 (Public Review)]

Summary:

The authors present a detailed study of a nearly complete Entomophthora muscae genome assembly and annotation, along with comparative analyses among related and non-related entomopathogenic fungi. The genome is one of the largest fungal genomes sequenced, and the authors document the proliferation and evolution of transposons and presence/absence of related genetic machinery to explore how this may have occurred. There has also been an expansion in gene number, which appears to contain many "novel" genes unique to E. muscae. Functionally, the authors were interested in CAZymes, proteases, circadian clock related genes (due to entomopathogenicity/ host manipulation), other insect pathogen specific genes, and secondary metabolites. There are many interesting findings including expansions in trahalases, unique insulinase and another peptidase, and some evidence for RIP in Entomophthoralean fungi. The authors performed a separate study examining E. muscae species complex and related strains. Specifically, morphological traits were measured for strains and then compared to the 28S+ITS-based phylogeny, showing little informativeness of these morpho characters with high levels of overlap.

This work represents a big leap forward in genomics of non-Dikarya fungi and large fungal genomes. Most of the gene homologs have been studied in species that diverged hundreds of millions of years ago, and therefore using standard comparative genomic approaches are not trivial and still relatively little is known. This paper provides many new hypotheses and potential avenues of research about fungal genome size expansion, entomopathogenesis in zygomycetes, and cellular functions like RIP and circadian mechanisms.

Strengths:

There are many strengths to this study. It represents a massive amount of work and a very thorough functional analysis of the gene content in these fungi (which are largely unsequenced and definitely understudied). Too often comparative genomic work will focus on one aspect and leave the reader wondering about all the other ways genome(s) are unique or different from others. This study really dove in and explored the relevant aspects of the E. muscae genome.

The authors used both a priori and emergent properties to shape their analyses (by searching for specific genes of interest and by analyzing genes underrepresented, expanded, or unique to their chosen taxa), enabling a detailed review of the genomic architecture and content. Specifically, I'm impressed by the analysis of missing genes (pFAMs) in E. muscae, none of which are enriched in relatives, suggesting this fungus is really different not by gene loss, but by its gene expansions.

Analyzing species-level boundaries and the data underlying those (genetic or morphological) is not something frequently presented in comparative genomic studies, however, here it is a welcome addition as the target species of the study is part of a species complex where morphology can be misleading and genetic data is infrequently collected in conjunction with the morphological data.

Weaknesses:

The conclusions of this paper are well supported, and I think the clarifications and improvements made to the manuscript in the revision process have greatly improved the paper.

---

## [Author Response]

The following is the authors’ response to the original reviews.

**Public Reviews:**

**Reviewer #1 (Public Review):**
Summary:The authors present a detailed study of a nearly complete Entomophthora muscae genome assembly and annotation, along with comparative analyses among related and non-related entomopathogenic fungi. The genome is one of the largest fungal genomes sequenced, and the authors document the proliferation and evolution of transposons and the presence/absence of related genetic machinery to explore how this may have occurred. There has also been an expansion in gene number, which appears to contain many "novel" genes unique to E. muscae. Functionally, the authors were interested in CAZymes, proteases, circadian clock related genes (due to entomopathogenicity/ host manipulation), other insect pathogenspecific genes, and secondary metabolites. There are many interesting findings including expansions in trahalases, unique insulinase, and another peptidase, and some evidence for RIP in Entomophthoralean fungi. The authors performed a separate study examining E. muscae species complex and related strains. Specifically, morphological traits were measured for strains and then compared to the 28S+ITSbased phylogeny, showing little informativeness of these morpho characters with high levels of overlap.This work represents a big leap forward in the genomics of non-Dikarya fungi and large fungal genomes. Most of the gene homologs have been studied in species that diverged hundreds of millions of years ago, and therefore using standard comparative genomic approaches is not trivial and still relatively little is known. This paper provides many new hypotheses and potential avenues of research about fungal genome size expansion, entomopathogenesis in zygomycetes, and cellular functions like RIP and circadian mechanisms.Strengths:There are many strengths to this study. It represents a massive amount of work and a very thorough functional analysis of the gene content in these fungi (which are largely unsequenced and definitely understudied). Too often comparative genomic work will focus on one aspect and leave the reader wondering about all the other ways genome(s) are unique or different from others. This study really dove in and explored the relevant aspects of the E. muscae genome.The authors used both a priori and emergent properties to shape their analyses (by searching for specific genes of interest and by analyzing genes underrepresented, expanded, or unique to their chosen taxa), enabling a detailed review of the genomic architecture and content. Specifically, I'm impressed by the analysis of missing genes (pFAMs) in E. muscae, none of which are enriched in relatives, suggesting this fungus is really different not by gene loss, but by its gene expansions.Analyzing species-level boundaries and the data underlying those (genetic or morphological) is not something frequently presented in comparative genomic studies, however, here it is a welcome addition as the target species of the study is part of a species complex where morphology can be misleading and genetic data is infrequently collected in conjunction with the morphological data.

Thank you for your careful reading of our work. We’re glad that you identified these areas as strengths.

Weaknesses:The conclusions of this paper are mostly well supported by data, but a few points should be clarified.In the analysis of Orthogroups (OGs), the claim in the text is that E. muscae "has genes in multi-species OGs no more frequently than Enotomophaga maimaiga. (Fig. 3F)" I don't see that in 3F. But maybe I'm really missing something.

Thank you for catching this. You were, in fact, not missing anything at all. There was a mismatch between the data plotted in F and G and how the caption described these data. We very much apologize for the confusion that this must have caused. We have corrected these plots and also made changes to improve interpretability (see below).

Also related, based on what is written in the text of the OG section, I think portions of Figure 3G are incorrect/ duplicated. First, a general question, related to the first two portions of the graph. How do "Genes assigned to an OG" and "Genes not assigned to an OG" not equal 100% for each species? The graph as currently visualized does not show that. Then I think the bars in portion 3 "Genes in speciesspecific OG" are wrong because in the text it says "N. thromboides had just 16.3%" species-specific OGs, but the graph clearly shows that bar at around 50%. I think portion 3 is just a duplicate of the bars in portion 4 - they look exactly the same - and in addition, as stated in the text portion 4 "Potentially speciesspecific genes" should be the simple addition of the bars in portion 2 and portion 3 for each species.

As mentioned above, we sincerely regret the error made in the plot and for the confusion that this caused. F now reflects the percentage of orthogroups (OGs) that possess at least one representative from the indicated species (left) and the percentage of OGs that are species-specific (only possess genes from one species; right). The latter is a subset of the former. G now reflects the percentage of annotated genes that were assigned an OG, per species, as well as the inverse of this - genes that were not assigned to any OG. These should, and now do, sum to 100%. The “Within species-specific OG” data summed with the “Not assigned OG” data yields the “Potentially species-specific data” in the rightmost column.

In the introduction, there is a name for the phenomenon of "clinging to or biting the tops of plants," it's called summit disease. And just for some context for the readers, summit disease is well-documented in many of these taxa in the older literature, but it is often ignored in modern studies - even though it is a fascinating effect seen in many insect hosts, caused by many, many fungi, nematodes (!), etc. This phenomenon has evolved many times. Nice discussions of this in Evans 1989 and Roy et al. 2006 (both of whom cite much of the older literature).

You’re right. We have now clarified that this behavior is called “summit disease” and referenced the suggested articles, along with a more recent review.

**Reviewer #2 (Public Review):**
In their study, Stajich and co-authors present a new 1.03 Gb genome assembly for an isolate of the fungal insect parasite Entomophthora muscae (Entomophthoromycota phylum, isolated from *Drosophila* hydei). Many species of the Entomophthoromycota phylum are specialised insect pathogens with relatively large genomes for fungi, with interesting yet largely unexplored biology. The authors compare their new E. muscae assembly to those of other species in the Entomophthorales order and also more generally to other fungi. For that, they first focus on repetitive DNA (transposons) and show that Ty3 LTRs are highly abundant in the E. muscae genome and contribute to ~40% of the species' genome, a feature that is shared by closely related species in the Entomophthorales. Next, the authors describe the major differences in protein content between species in the genus, focusing on functional domains, namely protein families (pfam), carbohydrate-active enzymes, and peptidases. They highlight several protein families that are overrepresented/underrepresented in the E. muscae genome and otherEntomophthorales genomes. The authors also highlight differences in components of the circadian rhythm, which might be relevant to the biology of these insect-infecting fungi. To gain further insights into E. muscae specificities, the authors identify orthologous proteins among four Entomophthorales species. Consistently with a larger genome and protein set in E. muscae, they find that 21% of the 17,111 orthogroups are specific to the species. To finish, the authors examine the consistency between methods for species delineation in the genus using molecular (ITS + 28S) or morphological data (# of nuclei per conidia + conidia size) and highlight major incongruences between the two.Although most of the methods applied in the frame of this study are appropriate with the scripts made available, I believe there are some major discrepancies in the datasets that are compared which could undermine most of the results/conclusions. More precisely, most of the results are based on the comparison of protein family content between four Entomophthorales species. As the authors mention on page 5, genome (transcriptome) assembly and further annotation procedures can strongly influence gene discovery. Here, the authors re-annotated two assemblies using their own methods and recovered between 30 and 60% more genes than in the original dataset, but if I understand it correctly, they perform all downstream comparative analyses using the original annotations. Given the focus on E. muscae and the small sample size (four genomes compared), I believe performing the comparisons on the newly annotated assemblies would be more rigorous for making any claim on gene family variation.

Thank you for this comment. While we did compare gene model predictions for two of these assemblies to assess if this difference could account for discrepancies in gene counts, completely reannotating all non-E. muscae datasets was outside of the scope of this study. In our opinion, the total number of predicted genes in a genome is not a best representation of differences since splitting or fusing gene models can inflate seeming differences; the orthology and domain counts are a more accurate assessment of the content. It’s possible that annotation differences may have inflated some gene family counts, however we will note that similar domain trends were observed between the closest species to E. muscae, Entomophaga maimaiga, suggesting that these differences were not sufficient to prevent us from detecting real biological signals. We look forward to continued improvement of our genome through additional sequencing and more clarity on total gene content of E. muscae.

The authors also investigate the putative impact of repeat-induced point mutation on the architecture of the large Entomophthorales genomes (for three of the eight species in Figure 1) and report low RIP-like dinucleotide signatures despite the presence of RID1 (a gene involved in the RIP process in *Neurospora crassa*) and RNAi machinery. They base their analysis on the presence of specific PFAM domains across the proteome of the three Entomophthorales species. In the case of RID1, the authors searched for a DNA methyltransferase domain (PF00145), however other proteins than RID1 bear such functional domain (DNMT family) so that in the current analysis it is impossible to say if the authors are actually looking at RID1 homologs (probably not, RID1 is monophyletic to the Ascomycota I believe). Similar comments apply to the analysis of components of the RNAi machinery. A more reliable alternative to the PFAM analysis would be to work with full protein sequences in addition to the functional domains.

While we understand this concern regarding domain vs. full length protein, the advantage of the domain search is that HMM-based searches are sensitive to detecting more distantly related homologs. Entomophthoralean fungi are distantly related from the ascomycetes in which these mechanisms have been characterized, so we chose a broader search approach that may identify proteins with similar domain structure, but are not necessarily homologs. These searches are presented in the manuscript as preliminary, but worth further investigation. However, our RID-based analysis did not identify convincing homologs for RID1 in entomophthoralean fungi included in our investigation, and we reported low homology (i.e., 12-14%) among our orthogroup of interest and RID1. We have further edited this section to clarify our understanding that these candidates are not RID1 homologs. We had hoped to avoid this implication, but we felt this investigation and null result were worth reporting.

**Recommendations for the authors:**

**Reviewer #1 (Recommendations For The Authors):**
Specific points:Results:"1.03 Gb genome consisting of 7,810 contigs (N50 = 301.1 kb). Additional... resulted in a final contig count of 7,810 (N50 = 329.6 kb)" So you started and ended with the same contig count but a different N50? Is this a typo?

Yes, this was a typo. Thank you for bringing this to our attention.

Figure 1D.The colors of Complete1x and Complete2x are too similar to tell them apart.

The colors have been made more distinct.

Figure 4B.I know C. rosea has been found from insects before, but it's mostly a mycoparasite and occasionally an endophyte, and has bioactivity against a lot of things. I just saw that it's listed as an entomopathogen, and I was surprised. Anyway, leave it as is if you want to, but it's definitely better studied and better known (Google Scholar) as a mycoparasite.

Thanks for this comment. For the sake of including a more diverse representation of entomopathogenic fungi, we have opted to leave this as is.

Full references (from the public comment)Evans, H.C., 1989. Mycopathogens of insects of epigeal and aerial habitats. Insect-fungus interactions, pp.205-238.Roy, H.E., Steinkraus, D.C., Eilenberg, J., Hajek, A.E. and Pell, J.K., 2006. Bizarre interactions and endgames: entomopathogenic fungi and their arthropod hosts. Annu. Rev. Entomol., 51, pp.331-357.
**Reviewer #2 (Recommendations For The Authors):**
I believe the manuscript could largely benefit from restructuring the results section to enhance clarity. The results section reads like a lot of descriptive back and forth, so that the reader lacks a clear rationale. The absence of a consistent dataset used for the different comparisons made all along the manuscript makes it hard to follow.Minor comments:(No line numbers were available so I refer to page numbers).p1not sure about the use of "allied" to describe other fungal species in the title and after (sister species?).

We didn’t want to use the word sister because not all of these species could be considered sister.

Genomic defence against transposable elements rather than "anti"?

We have rephrased to genomic defense.

p3Extra parenthesis at Bronski et al.

This is now corrected.

What does newly-available mean here?

We mean recent. A lot of the datasets we used were very new, and we wanted to emphasize that point.

The back and forth between genomes and transcriptomes makes it hard to follow, would clarify from the beginning (in addition to the sequencing method - short vs long-read assemblies as in Figure 1B) or perhaps use a consistent dataset for all subsequent comparative analysis in the Entomophthorales.

We have denoted our transcriptomic datasets in Fig 1C using parentheses.

p5Perhaps clarify that class II DNA transposons can also "copy" (single-strand excisions can be repaired by the host machinery).

We have now included mention of “copy” as well as “jump” mechanisms of Class II transposons per your suggestion.

p6"beginning roughly concurrently", not clear what "began".

This is now corrected.

"control" rather than "protect against"?

We’ve changed “protect against” to “counter”.

I believe RIP has only been observed (experimentally) in a handful of fungal species, all from the Ascomycota phylum.

Hood et al, 2005 found signatures of RIP in anther-smut fungus and Horns et al, 2012, found evidence of hypermutability across repeat elements within several Pucciniales species.

"RID1 contains two DNA_methylase domains", RID1 has one methyltransferase domain according to the reference Freitag et al, 2002.

Thank you for drawing this to our attention. It is true RID1 has one methyltransferase region; however, the sequence deposited by Freitag et al, 2002 (AAM27408) is predicted by HMMer to have two adjacent Pfam DNA_methylase domains (i.e., PF00145). In this exploratory analysis, we tried to leverage this characteristic to identify candidate proteins of interest. We have reworded this section to clarify this.

p8Here and after I would use more informative titles for each paragraph.

With the exception of the headings for Pfam, CAZy and MEROPs analyses, we believe the other headings are informative. We appreciate this comment, but opt to leave the heading titles as is.

I believe presenting the orthology analysis before the more in-depth protein family domain search.

We leveraged the OG analysis mostly as a way to identify potentially unique genes in E. muscae, so we think the current order makes the most sense.

p10Figures 3F and G are confusing. The legend for Figure 3F mentions "OGs with >= 2 species" while the figure shows "multi-species OGs", and reads as redundant with the "species-specific" OGs. For the "OGs within species" do I understand it correctly that it represents the number of genes assigned to OGs for each species? If yes, the numbers are in contradiction with Figure 3G. And in Figure 3G shouldn't the sum of "genes assigned in OGs" and "genes nor assigned in OGs" add up to 100? I'm probably missing something here, but I would clarify what the different sets of orthogroups are in the figure and in the text (perhaps adopting a pangenome-like nomenclature).

Thanks for this comment. This legend, unfortunately, reflected an earlier version of the figure and was overlooked prior to submission. We have since amended this and sincerely apologize for the error on our part.

p12The whole first paragraph reads more like it should be part of an introduction/discussion.

We’ve moved some of this paragraph to the discussion but left the background information necessary for the reader to understand why we were looking for homologs of wc and frq.

p13The last paragraph reads like discussion.

We have revised this paragraph so it now reads: “Because E. muscae is an obligate insect-pathogen only living inside live flies, we investigate the presence of canonical entomopathogenic enzymes in the genome. We find that E. muscae appear to have an expanded group of acid-trehalases compared to other entomopathogenic and non-entomopathogenic Entomophthorales (Fig. 4A), which correlates with the primary sugar in insect blood (hemolymph) being trehalose (Thompson, 2003). The obligate insectpathogenic lifestyle is also evident when comparing the repertoire of lipases, subtilisin-like serine proteases, trypsins, and chitinases in our focal species versus Zoopagomycota and Ascomycota fungi that are not obligate insect pathogens (Fig. 4B). Sordariomycetes within Ascomycota contains the other major transition to insect-pathogenicity within the kingdom Fungi (Araújo and Hughes, 2016). Based on our comparison of gene numbers, Entomophthorales possess more enzymes suitable for cuticle penetration than Sordariomycetes (Fig. 4B). In contrast, insect-pathogenic fungi within Hypocreales possess a more diverse secondary metabolite biosynthesis machinery as evidenced by the absence of polyketide synthase (PKS) and indole pathways in Entomophthorales (Fig. 4C).”

p15 and 16This all reads as redundant with the previous protein family domain analysis. I would try to merge them.

Thank you for this comment, however we have opted to maintain the current structure.

p18In the first sentence, I'm not sure about what was performed here.

This has been reworded to clarify.

p20Regarding the assembly, do I understand it correctly that a nuclear genome can be partially haploid / diploid?

Thanks for your comment. The genome itself is, of course, some integer multiple of n, but based onBUSCO scores our assembly doesn’t appear to have completely collapsed into a haploid genome. We think it makes more sense here to say “partially haploid” than “partially diploid” so have altered this.

p21RIP has only been observed in a couple of Ascomycetes. RIP-like genomic signatures (GC bias) have been observed elsewhere.

Hood et al, 2005 found signatures of RIP in anther-smut fungus and Horns et al, 2012, found evidence of hypermutability across repeat elements within several Pucciniales species.

p23Interesting that the peptidase A2B domain is found uniquely in E. muscae genome and is associated with Ty3 activity. Does the domain often overlap with annotated Ty3 in E. muscae genome? Or how come the domain is not present in other sister species with large genomes full of Ty3 transposons? Could it relate to a new active transposon in E. muscae specifically?

Thanks for this comment. The domain-based analysis was only performed on the predicted transcriptome of the genome assembly, which does not include the repeat elements (e.g., Ty3). It could be that this peptidase reflects a new active transposon that’s specific to E. muscae, which would certainly be very interesting. We’ve now included this idea in the discussion.

p26In the case of fungal genomes, I would not advise masking the assembly for repeated sequences prior to gene annotation (in particular given the current focus on protein family variation).

Thank you for this comment, however we disagree with this assertion as a typical approach for genome annotation in fungi and eukaryotic genomes is to use soft masking of transposable elements before performing gene prediction to avoid over-prediction. While there could be alternative approaches that compare masked or unmasked. This is a recommended protocol for underlying tools like Augustus (10.1002/cpbi.57) and in general descriptions of genome annotation (10.1002/0471250953.bi0401s52). The false positive rate of genes predicted through TE regions is likely to be more a problem than false negatives of missed genes in our experience. Further it seems appropriate to use consistent approach to annotation throughout when including genomes from other sources (e.g., Joint Genome Institute annotated genomes) which also use a repeat masking approach first before annotation. It seems most appropriate to use consistent methods when generating datasets to be used for comparative analyses. It is outside the scope of this project to reannotate all genomes with and without repeat masking.

p27Interrupted sentence at "Classification of DNA and LTR .. by similarity The".

This was an unnecessary partial phrase as the information on classification of elements via RepBase was made a few sentences above this.

p28Enriched/depleted rather than "significantly different"?

Thank you for this comment, however we have opted to maintain the current phrasing.